# Heterogeneous Ice Nucleation Properties of Natural Desert Dust Particles Coated with a Surrogate of Secondary Organic Aerosol

Zamin A. Kanji[1], Ryan C. Sullivan[2,3], Monika Niemand[4], Paul J. DeMott[2], Anthony J. Prenni[2,5], Cedric Chou[1], Harald Saathoff[4] and Ottmar Möhler[4]

[1]Department of Environmental System Sciences, Institute for Atmospheric and Climate Science, ETH Zürich, 8092, Zurich, Switzerland
[2]Department of Atmospheric Science, Colorado State University, Fort Collins, CO 80523, USA
[3]Center for Atmospheric Particle Studies, Carnegie Mellon University, Pittsburgh, PA 15213, USA
[4]Institute of Meteorology and Climate Research, Karlsruhe Institute of Technology, 76344, Eggenstein-Leopoldshafen, Germany
[5]Now at Air Resources Division, National Park Service, Lakewood, CO 80228, USA

*Correspondence to*: Zamin. A. Kanji (zamin.kanji@env.ethz.ch)

## Abstract

Ice nucleation abilities of surface collected mineral dust particles from the Sahara (SD) and Asia (AD) are investigated for the temperature ($T$) range 253 – 233 K and for supersaturated relative humidity ($RH$) conditions in the immersion-freezing regime. The dust particles were also coated with a proxy of secondary organic aerosol (SOA) from the dark ozonolysis of $\alpha$-pinene to better understand the influence of atmospheric coatings on the immersion freezing ability of mineral dust particles. The measurements are conducted on poly-disperse particles in the size range 0.01 – 3 µm with three different ice nucleation chambers. Two of the chambers follow the continuous flow diffusion chamber (CFDC) principle (Portable Ice Nucleation Chamber, PINC) and the Colorado State University CFDC (CSU-CFDC), whereas the third was the Aerosol Interactions and Dynamics in the Atmosphere (AIDA) cloud expansion chamber. From observed activated fractions ($AF$) and ice nucleation active site ($INAS$) densities, it is concluded within experimental uncertainties that there is no significant difference between the ice nucleation ability of the particular SD and AD samples examined. A small bias towards higher $INAS$ densities for uncoated versus SOA coated dusts is found but this is well within the 1$\sigma$ (66 % prediction bands) region of the average fit to the data, which captures 75 % of the $INAS$ densities observed in this study. Furthermore, no systematic differences are observed between SOA coated and uncoated dusts in both SD and AD cases, regardless of coating thickness (3 – 60 nm). The results suggest that any differences observed are within the uncertainty of the measurements or differences in cloud chamber parameters such as size fraction of particles sampled, and residence time, as well as assumptions in using $INAS$ densities to compare poly-disperse aerosol measurements which may show variable composition with particle size. Coatings with similar properties to that of the SOA in this work and with coating thickness up to 60 nm are not expected to impede or enhance the immersion mode ice nucleation ability of mineral dust particles.

## 1. Introduction

Ice nucleation in mixed-phase clouds (MPCs) is an important process that can significantly modify cloud microstructure causing glaciation and initiating precipitation thus impacting cloud albedo, lifetime and radiative properties. In the absence of ice crystals falling into the clouds from above-lying cloud layers, primary ice formation via heterogeneous freezing of aerosol particles at temperatures above 235 K is responsible for ice formation in MPCs. After primary ice formation, MPCs can fully glaciate due to secondary ice formation. Glaciation can also occur due to the co-existence of ice and liquid in the same environment causing ice crystals to grow at the expense of liquid droplets via the Bergeron-Wegner-Findeisen process (Korolev, 2007). It is therefore important to quantify heterogeneous ice nucleation relevant to MPC temperatures to predict primary ice formation and subsequent secondary ice formation processes.

Contact and immersion freezing are the two known heterogeneous ice nucleation processes thought to be relevant to MPC formation conditions because they involve freezing of supercooled liquid water droplets. Contact freezing involves freezing initiated at the surface of supercooled droplets upon collision with aerosol particles. In immersion freezing, a supercooled droplet formed on an insoluble aerosol particle – called an ice nucleating particle (INP) – will initiate freezing at the interface of the INP surface and the water droplet upon sufficient supercooling. A further distinction is made between immersion and condensation freezing. In immersion freezing, a droplet is formed at higher temperatures and undergoes supercooling before freezing, whereas in condensation freezing, the liquid water condensation occurs at supercooled temperatures and freezing is thought to be initiated at the same temperature during the liquid water condensation process or immediately thereafter.

A number of atmospheric particles have been shown to induce heterogeneous ice nucleation in the temperature regime above 235 K for a wide range in *RH* (Kanji et al., 2017). These can range from particles of biological origin associated with marine particles (Wilson et al., 2015; DeMott et al., 2016; Ladino et al., 2016; McCluskey et al., 2016), to those associated with terrestrial particles such as agricultural soil dust and leaf litter (Huffman et al., 2013; Prenni et al., 2013; Hill et al., 2014; Tobo et al., 2014; Steinke et al., 2016), desert dust (DeMott et al., 2010; Hoose and Möhler, 2012; DeMott et al., 2015) and organic aerosol (DeMott et al., 2003a; Wang et al., 2012a; Knopf et al., 2014; Knopf et al., 2018). The impact of these particles on ice formation processes in the troposphere is only quantifiable if their spatiotemporal distributions and burdens are well known. Mineral dust particles of desert origin have a significant tropospheric burden, and role in heterogeneous ice nucleation in both laboratory experiments (Hoose and Möhler, 2012; Murray et al., 2012) and by inference from atmospheric observations (DeMott et al., 2003a; DeMott et al., 2003b; Chou et al., 2011; Cziczo et al., 2013; Boose et al., 2016a).

Mineral dust particles frequently accumulate secondary components such as acids during atmospheric transport (Bates et al., 2004; Arimoto et al., 2006; Sullivan et al., 2007a; Sullivan et al., 2007b; Sullivan and Prather, 2007; Shi et al., 2008; Tobo et al., 2009). Observations of the ice nucleation activity of mineral dust coated with soluble material (mostly inorganic acids) are

used as proxies for particles that undergo processes of atmospheric ageing during long-range transport. Varying degrees of suppression (none, partial, or complete) in the ice nucleation activity of mineral dust particles due to in/organic coatings or exposure to inorganic acid vapours has been observed and is dependent on the $RH$ and $T$ regimes or chemical processing type of the dust particles (Archuleta et al., 2005; Möhler et al., 2008; Reitz et al., 2011; Tobo et al., 2012; Augustin-Bauditz et al.,

2014; Wex et al., 2014; Freedman, 2015). Sullivan et al. (2010a and 2010b) showed that at 243 K for $RH$ with respect to water ($RH_w$) < 100% in the deposition regime, $HNO_3$ and $H_2SO_4$ coated Arizona Test Dust (ATD) particles exhibited lower activated fractions ($AF$) close to the detection limit – implying complete deactivation of ice nucleation activity – compared to uncoated ATD particles, a similar conclusion later reported by Kulkarni et al. (2014, 2015). However, for ice formation conditions at $RH_w$ > 100% corresponding to the immersion freezing regime, the $HNO_3$ coated particles yielded similar $AF$ to uncoated ATD

(Sullivan et al., 2010a; Kulkarni et al., 2015) whereas the $H_2SO_4$ coated ATD exhibited lower $AF$s (partial deactivation) compared to the uncoated ATD in Sullivan et al. (2010b) but not in (Kulkarni et al., 2014). A complete suppression of ice nucleation abilities in the immersion mode at 247 and 243 K and no supression at 239 K for kaolinite particles coated with $H_2SO_4$ was also reported by Tobo et al. (2012). The response of ice nucleation to coatings below versus above water saturation was attributed to the dissolution of $HNO_3$ and its reaction products from the dust particle surface following droplet activation

at $RH_w \geq 100\%$, thus restoring the ice active sites for immersion freezing that had been impaired for deposition nucleation by the acid uptake (Sullivan et al., 2010a; Sullivan et al., 2010b; Niedermeier et al., 2011a). In the case of the $H_2SO_4$ coating, it was concluded that the ice active surface sites of the ATD were irreversibly chemically and probably morphologically modified thus causing a reduction in the observed ice nucleation ability (Niedermeier et al., 2011a, b; Reitz et al., 2011) via all mechanisms. An alternative explanation is that $H_2SO_4$ can induce different chemical reactions with components of Arizona

test dust than $HNO_3$ can, and/or that the reaction products from $H_2SO_4$ chemistry are not dissolved off the mineral surface to restore the active sites after droplet activation (Sullivan et al., 2010b). Furthermore, for water supersaturated conditions and temperatures between 248 – 238 K, Kulkarni et al. (2014, 2015) also report no influence of $H_2SO_4$ and $HNO_3$ coatings regardless of coating thickness (between 1 - 40 nm) on the ice nucleation activity of ATD, illite, feldspar and montmorillotnite particles despite showing that the surface structural order of the coated particles had been modified. The authors concluded

that products formed between the reactions of $H_2SO_4$ and the substrate aerosol (e.g. $Al_2(SO_4)_3$ (Panda et al., 2010; Sihvonen et al., 2014)) would dissolve in the immersion freezing regime, exposing the modified aerosol substrate active sites for ice nucleation. A suppression of ice nucleation activity for $RH_w$ < 100% by $H_2SO_4$ treated kaolinite and montmorillonite particles between 235 – 245 K has also been reported by Sihvonen et al. (2014) and Chernoff and Bertram (2010), with the latter study showing a higher degree of suppression (higher $RH$ required for ice nucleation). The degree of suppression in ice nucleation

activity is likely related to the acid concentration and contact time with the particles prior to ice nucleation, with longer times and higher acid concentration leading to greater deactivation of the ice-nucleating sample (Sihvonen et al., 2014). The suppression in ice nucleation activity due to $H_2SO_4$ coatings were linked to the formation of hydrated $Al_2(SO_4)_3$ ($Al_2(SO_4)_3.17H_2O$) products which surrounds the clay particles, and deliquesce before freezing (at 240 and 245 K), or nucleate ice via deposition at the coldest temperature investigated (235 K). Sihvonen et al. (2014) examined the same clay samples

treated with $HNO_3$ for their ice nucleation ability at the same $RH_w$ and temperature range. They observed a suppression in ice nucleation activity for kaolinite but not for montmorillonite and related this to the absence of the formation of a new product on the montmorillonite particles, despite its structure changing as observed through XRD measurements in agreement with (Kulkarni et al., 2014, 2015). Lastly, the partial deactivation of kaolinite and ATD particles when exposed to high (ppm level) concentrations of $O_3$ have also been reported in both the deposition and immersion mode (Kanji et al., 2013). On the other hand, an enhancement in immersion freezing is reported requiring warmer temperatures for freezing when trace amounts of $(NH_4)_2SO_4$ are added to aqueous droplets containing feldspar (microcline) particles compared to pure water droplets (Kumar et al., 2018; Whale et al., 2018). Similarly, enhanced immersion freezing was observed for kaolinite particles exposed to low $O_3$ (ppb level) but no difference was observed for the same $O_3$ exposure of ATD particles (Kanji et al., 2013). The discussion of results of ageing and coating expeirments above strongly suggests that the type of substrate aerosol, coating or ageing component and regime of ice nucleation (water sub- or supersaturated) are all important considerations when assessing impact of coatings or ageing on the ice nucleation activity of particles. It cannot be assumed that a coating or chemical processing will necessarily impair ice nucleation properties of mineral dust, in particular in the immersion freezing mode.

In the atmosphere, organic aerosol particles that are internally or externally mixed with other aerosol have also been identified as INPs, in particular, under $T$ and $RH$ conditions relevant for cirrus cloud formation (DeMott et al., 2003a; Richardson et al., 2007; Baustian et al., 2012; Knopf et al., 2018). Offline freezing analysis of atmospheric aerosol composed mostly of anthropogenic organics sampled onto substrates also showed ice nucleation activity in the immersion and deposition regimes. Probing of the specific INPs at the onset of ice initiation conditions revealed that compositions of particulates almost entirely organic in nature (Knopf et al., 2010) with no unique structural or compositional features (Wang et al., 2012b; Knopf et al., 2014). In laboratory studies organic glassy aerosol composed of raffinose, levoglucosan or citric acid were demonstrated to be INPs in the cirrus regime (Murray et al., 2010; Wilson et al., 2012) whereas oxalic acid, succinic acid, and multicomponent mixtures with $(NH_4)_2SO_4$ have been demonstrated to be effective INPs in the cirrus and MPC temperature regimes (Wagner et al., 2011; Wagner et al., 2012; Wagner et al., 2014; Wagner et al., 2015). Studies of ice nucleation of secondary organic aerosol (SOA) surrogates from the reaction products of naphthalene and OH radicals (Wang et al., 2012a) found such particles to be effective at forming ice in both the immersion (for $T > 230$ K, at $RH <$ required for homogeneous freezing) and deposition mode (for $T < 230$ K). However, another study that looked at ice nucleation of $\alpha$-pinene SOA and its reaction products with OH radicals (Ladino et al., 2014) found both sets of particles to be ineffective INPs requiring homogeneous freezing conditions to observe ice nucleation. Similar conclusions were reached for the ice nucleation ability of SOA produced from ozonolysis of 25 different alkene precursors (Prenni et al., 2009). Ladino et al. (2014) concluded that the detailed composition is not of paramount importance to the ice nucleating abilities of the SOA particles, however particle conditioning such as precooling or pre-activation improved the observed ice nucleation ability. Given that the SOA particles studied by Wang et al. (2012a) acted as INPs, it may be that the parent particle composition or an experimental technique that allows appreciable cooling of aerosols prior to ice nucleation aids the ability of the aerosol to act as an INP. While overall the efficiency of organic aerosol to nucleate

ice remains lower than that of mineral dust, the role of non-mineral aerosol in ice nucleation could be important in regions of the troposphere where mineral dust is scarce or absent.

The ubiquity of organic aerosol and the long residence times of dust in the troposphere can result in organic-mineral internally mixed particles (Russell et al., 2002; Maria et al., 2004; Sullivan et al., 2007a; Sullivan and Prather, 2007; Li et al., 2014). The ice nucleation properties of organic aerosol coatings on mineral dust as a proxy for such internal mixtures are therefore of interest to better understand the atmospheric relevance of ice nucleation of mineral aerosols. Tobo et al. (2012) reported that for kaolnite particles coated with levoglucosan (unreactive with kaolinite), deposition mode nucleation was suppressed (for $T$ > 239 K), but no suppression was observed for immersion mode ice nucleation. A deposition ice nucleation mode study by Möhler et al. (2008) showed significantly lower ice nucleation abilities for ATD and the clay mineral, illite coated with the oxidation products of $\alpha$-pinene and $O_3$. A $RH$ with respect to ice ($RH_i$) of between 10-50% higher was required for coated particles to nucleate ice compared to uncoated particles in the temperature range $205 - 210$ K. This suggests that SOA coatings can alter the ice nucleating properties of freshly emitted mineral dust aerosol at water sub-saturated conditions and temperatures relevant for the cirrus regime. Other than the study for levoglucosan coated kaolinite (Tobo et al., 2012) there are no other laboratory studies reporting ice nucleation properties of organic coated natural mineral dust in the MPC temperature regime ($T$ > 233 K and $RH_w \geq 100\%$), however a number of studies have been performed on particles composed of inorganic salts mixed with a variety of organic species (Parsons et al., 2004; Shilling et al., 2006; Wise et al., 2010; Knopf and Forrester, 2011) or on pure and multicomponent organic species (Fukuta, 1966; Prenni et al., 2009; Schill et al., 2014; Ignatius et al., 2016; Qiu et al., 2017). As mentioned above, the study by Möhler et al. (2008) explored only lower temperatures and water sub-saturated conditions. In this work, we present the ice nucleation properties of natural desert dust samples coated with the reaction products of $\alpha$-pinene and $O_3$ as a proxy for secondary organic aerosol at $RH_w$ > 100% and 253 K > $T$ > 233 K relevant to the MPC regime. To simulate atmospheric immersion freezing, we report results for conditions of supersaturation with respect to water, i.e., at thermodynamic conditions favourable for liquid water drops and ice crystals to form. It has been shown that operating continuous flow diffusion chambers (CFDCs), as is used in this work, at sufficiently high $RH_w$, allows one to estimate the maximum number of INPs that can activate at that temperature under the immersion freezing regime (DeMott et al., 2015). For example, activated fractions of dust INPs measured in a CFDC at $RH_w$ = 105% were observed to be within a factor of 2 to 3 of the frozen fractions observed in immersion freezing measured in cloud chamber expansions (DeMott et al., 2015).

## 2. Experimental Methods

The experiments described here were all conducted at the Institute of Meteorology and Climate Research at the Karlsruhe Institute of Technology in Karlsruhe, Germany as part of the third Aerosol Cloud Interaction (ACI-03) campaign. Two CFDCs and the Aerosol Interactions and Dynamics in the Atmosphere (AIDA) expansion chamber were used for INP measurements.

An aerosol preparation and characterization chamber (APC) with a volume of 3.7 m³ was used to suspend and prepare SOA coated and uncoated desert dust particles. In this section, we provide a description of the methods and instruments used to measure the ice nucleation abilities of the coated and uncoated desert dust particles. The INP counters, PINC (Portable Ice Nucleation Chamber, see section 3.2) and CSU-CFDC (Colorado State University – Continuous Flow Diffusion Chamber, see section 3.3) both had aerosol sampling ports to access the APC chamber as shown in Figure 1. The AIDA chamber was also filled with aerosol either directly from the aerosol generators or from the APC prior to cloud expansion experiments (see Figure 1).

## 2.1. Aerosol Processing and Sampling

In this work, the ice nucleation properties of two different desert dusts samples were examined, Asian Dust (AD) and Saharan Dust (SD). Samples from these regions were chosen because emissions of dust from arid and semi-arid regions mainly come from the Saharan and Asian regions (Tang et al., 2016 and references therein). The AD sample was collected off the surface in the eastern part of the Taklamakan Desert in China (east of the Dalimu basin between Kuerle and Ruoquiang) while the SD soil sample was collected from a hole 1.5 m deep about 70 km north-east of Cairo city, Egypt (Ullrich et al., 2017). Both the SD and AD samples contain low amounts of total soluble ionic species (0.63 and 0.45 % by weight, Megahed, 2007) suggesting that there should be minimal influence of soluble matter nor anthropogenic pollutants on the dust. The mineralogy of the dust particles have been investigated before for AD in the size fraction < 32 µm (called Taklamakan in Boose et al., 2016b) and SD in the size fraction of < 20 µm (labelled Cairo2 in Linke et al., 2006). Surface sampling was conducted for AD and SD so as to obtain enough sample mass to perform multiple ice nucleation experiments of coated (varying thickness) and uncoated dust samples using all three techniques presented here (see section 2.3-2.4).

For aerosol generation, the dust samples were sieved to obtain a sub-fraction with diameters < 75 µm. Samples were introduced into the APC using a rotating brush generator (RGB – 1000, Palas, GmbH, Germany) which is operated with dry high-purity synthetic air. Aerosol particle counts and number size distributions in the APC chamber were monitored by a condensation particle counter (CPC 3010, TSI Inc.) and scanning mobility particle sizer (SMPS 3080 TSI Inc.) for particle sizes with mobility diameters in the range 14 – 820 nm. In addition, an aerodynamic particle sizer (APS 3221, TSI Inc.) measured number size distributions of particles with aerodynamic diameters in the range 0.5 – 20 µm.

The SOA coating on the dust was applied at room temperature in the APC chamber by the *in-situ* ozonolysis of α-pinene. The addition of ozone to the dark chamber was always in excess to the amount of α-pinene added to ensure complete reaction of the hydrocarbon to SOA products (Möhler et al., 2008). The SOA yield with respect to mass (µg m⁻³) of α-pinene reacted is determined according to the parameterization described in the work of Saathoff et al. (2009) which describes the formation of SOA from the ozonolysis of α-pinene in the temperature range 233 – 313 K. The SOA yield is mainly a function of the temperature and the amount of α-pinene added to the APC chamber. The variation with temperature is not of much relevance

to the production of SOA in the current work given that the SOA coatings were always generated at room temperature in the APC chamber. Considering the transport of the gas-phase SOA products to the dust particles, the condensation of the SOA vapours onto the desert dust particles occurs in the transition regime from the kinetic to the continuum regime described by the Knudsen number. In the kinetic regime, for particles smaller than the gas mean free path, the diameter growth rate of the

SOA layer is influenced by the rate of random molecular collisions of vapour molecules with the particles and is independent of the particle size (Niemand, 2012). On the other hand, for particles larger than 0.5 – 1 µm the growth rate becomes inversely proportional to the radius (continuum regime). It is therefore assumed that the coating thickness decreases with increasing particle size.

Representative dust particle number and surface size distributions are shown in Figure 2 and 3 respectively for AD and SD. To convert mobility and aerodynamic diameters to volume equivalent diameters, a shape factor of 1.4 for AD (1.2 for SD) and a particle density of 2.6 g cm$^{-3}$ were used (Hiranuma et al., 2015). To determine the coating thickness, the total SOA produced is distributed over the surface area derived from the combined SMPS and APS measurements. In cases when a SOA only mode is formed as shown in Figure 2 and 3, the mass of the SOA mode is subtracted (assuming spherical particles with density of

1.25 g cm$^{-3}$) from the SOA yield (Saathoff et al., 2009) before distributing the SOA mass over the dust surface area to determine a mean coating thickness.

**2.2. The ETH Portable Ice Nucleation Chamber (PINC)**

Here we present a brief overview of PINC which is based on the original principle of the CSU-CFDC INP counter presented in Rogers (1988). A detailed account of the construction, operation validation and uncertainties can be found in Chou et al.

(2011) and Kanji et al. (2013). PINC is a parallel plate vertically oriented CFDC with anodized inner aluminium walls separated by a distance of 10 mm. The inner chamber walls are coated with ice by flooding the chamber with water at 253 K for 20 seconds resulting in an ice coating thickness of ~ 200 µm. In the chamber an aerosol flow is layered between two particle-free sheath flows so as to maintain a focused known location of the aerosol sample flow. The total flow rate of 10 lpm is composed of 1 lpm aerosol flow and two 4.5 lpm sheath flows to make a total sheath flow of 9 lpm. The chamber is made up of a growth

and evaporation section. Particles enter the growth region where a supersaturation with respect to ice and water are attained by increasing the temperature difference between the two ice coated walls. An increase in $RH$ is possible while keeping the sample temperature constant. Particle residence time in the growth region is ~ 4 to 5 seconds for the temperature regime studied here. After the growth region, particles pass into the evaporation section that is maintained at the same temperature as that of the warm wall thus ensuring particles are exposed to ice saturation conditions corresponding to $RH_w < 100\%$, allowing for

unfrozen liquid drops to shrink in size by evaporation. The residence time of the particles in this section is ~1-2 seconds before being detected by the optical particle counter (OPC, CliMET, 3100) which is mounted downstream of the evaporation section. Particles larger than 3 µm (optical diameter) are classified as ice crystals. To avoid misclassification of aerosol particles as ice crystals, particles enter the growth region through an impactor with a cut size of 0.7 µm (volume equivalent diameter

determined using shape factor and density mentioned above). A typical experiment consists of sampling aerosol continuously from the APC chamber while changing the $RH$ of the sample lamina from ice saturation to above water saturation to a maximum $RH_w$ above which droplets survive the evaporation region (water drop survival $RH$) upstream of being detected. Such a $RH$ ramp takes ~25 minutes resulting in an ice activation spectrum at a constant nominal sample temperature. Above

the water drop survival $RH_w$, ice crystals can no longer be distinguished from droplets by size discrimination in the OPC. In PINC for temperatures $\leq$ 253 K the water drop survival region occurs at $RH_w \geq 107\%$ and therefore all measurements are reported for $RH_w = 105\%$ to avoid signal contamination from unfrozen droplets when detecting ice in the OPC. The temperature measurement uncertainty in PINC is $\pm0.1$ K with a variability of temperature in the aerosol lamina of ~ 0.6 K. Uncertainty in reported $RH$ mostly due to temperature uncertainties are on the order of $\pm2\%$ $RH_w$ (absolute) for the temperature range reported

in this work. Relative uncertainties in CPC and OPC measurements are on the order of 10% each leading to uncertainty in activated fraction of ~15%.

### 2.3. The CSU-CFDC

INP measurements were also conducted with the CSU-CFDC model 1H whose basic description follows that provided in Rogers et al. (2001) with updates and modifications reported in  Petters et al. (2009) and Eidhammer et al. (2010). Briefly, the

CFDC consists of two (inner and outer) concentric copper cylindrical walls, oriented vertically, that form an annular gap of about 1.1 cm in which the laminar aerosol flow (1.5 lpm) sampled from the APC chamber is sandwiched in between particle-free sheath flows (8.5 lpm, divided on either side of the sample lamina). The inner and outer walls of the CSU-CFDC are coated with ~ 100 µm ice layers by pumping water through the chamber at 248 K. The temperatures of the inner and outer walls along with the position of the sample layer determine the INP processing temperature and $RH$ of the samples investigated.

The residence time of the aerosols at a given experimental processing temperature is approximately 4-5 seconds. The inner and outer walls of the lower third of the instrument are actively cooled and maintained at the colder (inner) wall temperature in order to drive conditions toward ice saturation so that particles that activated as droplets are evaporated in the exit region. This allows for clear optical detection of activated ice crystals. Particles that grow to sizes with optical diameters > 3 µm are counted as ice crystals. In order to ensure that large particles are not sampled at the inlet of the chamber, an impactor with an

aerodynamic particle cut off size of 1.5 µm (volume equivalent diameter of ~ 1 µm assuming densities and shape factors listed above for mineral dust) is used to filter out larger particles that could be miscounted as ice. If processing conditions in the chamber growth region remain at $RH_w < 108\%$ at most temperatures, droplets will evaporate below sizes that would be detected as ice by the OPC (Climet, CI – 3100) (DeMott et al., 2015). We therefore only report data that are processed at $RH_w = 105\%$. Measurements uncertainties in the CSU-CFDC vary with processing conditions, but are typically $\pm$ 0.5 K and $RH_w \pm 2.4\%$ at

243 K (DeMott et al., 2015). Particle and ice crystal counting errors to compute $AF$ are on the order of $\pm10\%$ or less.

## 2.4. AIDA Expansion Chamber

The AIDA cloud expansion chamber operates on the principle of volume expansions that result in rapid cooling thus increasing the $RH$ inside a 84 m$^3$ cylindrical chamber (Möhler et al., 2006). Details of the exact set up used in the current work are reported in Niemand et al. (2012) and Wagner et al. (2012), however we present a brief overview here. The AIDA chamber can be evacuated allowing expansion experiments with well-controlled and repeatable cooling rates of between 0.1 – 6 K min$^{-1}$ corresponding to updraft velocities of about 0.15 m s$^{-1}$ to 8 m s$^{-1}$ covering a range from weakly to strongly convective wave clouds. The chamber is housed in a thermally insulated container that can be cooled down to 183 K with homogeneous temperature control < ±0.3 K. Typical background aerosol concentrations in the chamber are less than 0.1 cm$^{-3}$. The chamber is evacuated to pressures as low as 0.01 hPa before an expansion experiment and refilled with synthetic air to ensure clean conditions. Before a cloud expansion experiment, constant temperature and pressure conditions in the chamber maintain the $RH_i$ between 90 – 95% by means of a thin ice coating on the inner walls of the AIDA chamber. At these conditions, the spatial and temporal temperature fluctuations are less than ±0.2 K, achieved by operating a mixing fan at the bottom of the chamber to ensure homogeneous conditions during experiments. Before an ice nucleation experiment is performed, a reference expansion where background particles are forced to activate as cloud droplets or ice crystals is performed which ensures precipitation of background particles from the chamber. A fresh sample of SOA-coated or uncoated dust is then transferred into the cloud chamber either directly from the aerosol generator or from the APC chamber. A tunable diode laser (TDL) is used to determine the water vapour pressure *in situ* with an accuracy of 5% (Niemand et al., 2012). In addition, total water concentration measurements were taken with a chilled-mirror hygrometer (373LX, MBW Calibration Ltd.). In cloud-free conditions, this compared to the TDL measurement to within 1 – 2% (Fahey et al., 2014). Ice crystals and water droplets were detected with two welas OPCs (Palas, GmbH, Germany) (Benz et al., 2005). Ice crystals were distinguished from interstitial aerosol and water droplets by their size since ice crystals grow to much larger sizes at the expense of liquid droplets due to the lower equilibrium vapour pressure of ice compared to supercooled water. In addition, for the optical arrangement of the welas OPC, frozen droplets and ice crystals have much larger scattering intensity than liquid droplets of the same volume. The size threshold used to count ice crystals was variable and set for each experiment such that all counts from interstitial aerosol and water droplets were below this threshold. The aerosol particle number concentration in AIDA was measured with a CPC (3010, TSI Inc.) which was modified and calibrated to take continuous measurements at pressures from 100 – 1000 hPa. Number size distributions were taken with an APS (3221, TSI Inc.) and SMPS (3080, TSI Inc.) before every cloud expansion experiment. The counting uncertainty for aerosol number and ice crystal measurements for AIDA is ± 20% each, resulting in a total uncertainty of ± 28% in the $AF$.

## 2.5. Active Fraction ($AF$) and Ice Active Surface Site Density ($INAS$) Determination

For all ice chambers, $AF$ is reported by determining the ratio of ice counts from the OPC to the aerosol counts derived from the relevant CPC. Furthermore, assuming spherical particles, the number size distributions were converted into surface

distributions that allowed for the derivation of ice active surface site densities, (*INAS*, Connolly et al., 2009; Hoose and Möhler, 2012; Murray et al., 2012; Beydoun et al., 2016). In determining the *INAS density* for experiments with PINC and CSU-CFDC we only considered surfaces areas of particles below 0.7 (1) and 1 (1.5) µm volume equivalent (aerodynamic) diameter, respectively, owing to the use of impactors upstream of the chambers. The same consideration was not applied to determine *AF* as the contribution of aerosol number above 1 µm (see Figure 2) is very small. However, the contribution to the surface area from the particles above 1 µm is significant enough (see Figure 3) to have to account for the impactors used upstream of PINC and CSU-CFDC in determining *INAS* densities.

## 3. Results and Discussion

AD and SD were tested for ice nucleation above water saturation (except at 233 K), in the immersion mode in the temperature range 253 to 233 K. For the CSU-CFDC and PINC experiments, ice nucleation abilities are reported for $RH_w = 105\%$ which is in the thermodynamic regime favouring condensation of water prior to or during freezing. As such these data could have contributions from deposition nucleation but given that we expect droplet formation during air entry and cooling to the processing temperature in the INP chambers at these conditions (Welti et al., 2014), the contributions from deposition nucleation are expected to be negligible. We choose $RH_w = 105\%$ as this is the upper limit of reliable operation for PINC at the warmest temperature reported here (253 K) and is commonly used in reporting ambient CFDC data (Sullivan et al., 2010a; Sullivan et al., 2010b; DeMott et al., 2017). To deduce the *AF* (and *INAS* densities) at $RH_w = 105\%$ in PINC and CSU-CFDC, the *AF* corresponding to $RH_w = 104 – 106\%$ were averaged thus each data point presented represents an average of 10-20 data points (recorded every 5 seconds for PINC and 1-second for CSU-CFDC) depending on the rate of change of $RH_w$ (1-2% min$^{-1}$). For the AIDA expansion, only those experiments where water drop formation was observed before ice formation are reported to ensure immersion freezing. Based on mineral dust experiments reported in DeMott et al. (2015) and correction factors presented in Garimella et al. (2017) we believe the *AF* to be lower than those in AIDA by at least a factor of ~ 3-4 primarily due to aerosol particles escaping the lamina resulting in being exposed to lower $RH_w$ values than the computed 105 %. To aid the comparison presented below, the *AF* values for PINC and CSU-CFDC have been increased by a factor of 3 based on the dust experiments calibration factor reported in DeMott et al. (2015) which used some of the experiments presented in this work.

### 3.1. Activated fractions of uncoated and coated AD and SD

Figure 4 and 5 show the *AF* increasing as a function of decreasing temperature for AD and SD respectively, a trend expected and observed by all three methods as more ice active sites become available at lower temperatures (see further discussion in section 4.2). A first observation is that there is no significant nor systematic difference at the $p < 0.05$ level (AD; *t*-value = 0.47, *p*-value = 0.32, SD; *t*-value = -1.38, *p*-value = 0.08) between the *AF* for coated and uncoated dusts in the immersion freezing regime for all ice chambers used in this work. Fit curves through the uncoated and coated data for both dusts with

95% confidence intervals (not shown) significantly overlap further supporting the lack of difference. There is an indication that at higher temperatures, $T > 245$ K, the CSU-CFDC observes lower $AF$ for the AD-SOA particles (experiment 5 and 6 for AD-SOA compared to 2 and 6 for AD). However, given that experiment 5 had a thicker SOA coating (~60 nm) and a higher $AF$ compared to experiment 6 with a coating of ~25 nm, suggests that the differences could be due to size, with larger particles

being more effective at activating to cloud droplets, solvating the coating, and freezing by immersion. For $T$ ~245 K no difference between coated and uncoated dust was observed in AIDA and furthermore for $235 < T < 245$ both PINC and CSU-CFDC observed no differences between the AD and AD-SOA. We thus conclude that in the immersion freezing regime, the SOA coating did not impact the ice nucleation ability of the AD samples tested here. This is in agreement with previous studies of $HNO_3$ (Sullivan et al., 2010a; Kulkarni et al., 2015) and $H_2SO_4$ (Kulkarni et al., 2014) coatings on ATD, illite, kaolinite,

montmorillonite and feldspars, all components of natural dust samples. For 233 K we report $AF$ for $RH_w$ between 95-98 % to exclude bias from homogenous freezing that could occur at $RH_w \geq 100\%$. We note that at ~ 233 K data from PINC suggest a small coating effect, with $AF$ of AD-SOA being slightly below that of AD. Depending on the mechanism of ice nucleation active here, this is not surprising given that similar coatings to that used here (Möhler et al., 2008) and inorganic acid coatings (Chernoff and Bertram, 2010; Sullivan et al., 2010a; Sullivan et al., 2010b; Tobo et al., 2012; Kulkarni et al., 2014; Sihvonen

et al., 2014; Kulkarni et al., 2015) were reported to impede deposition mode nucleation (i.e. at $RH_w < 100\%$). However, given the insignificance in overall coating effects (see above) and size biases (see discussion below), one cannot exclude that at such high $RH_w$ water uptake into the SOA coating could have led to complete or partial solvation of the coating leading to immersion freezing of the AD-SOA sample at 233 K.

Considering the results from the CSU-CFDC and AIDA chambers for all temperatures shown and for the $AF$ from PINC for $T < 245$ K, the SOA coating on the SD does not appear to impact its ice nucleation ability in the immersion mode regardless of coating thickness (see also Figure 6), similar to the conclusions from the AD results. At ~248 K there appears to be a major suppression in $AF$ for experiment 11 in PINC, where the SD-SOA has an $AF$ of three orders of magnitude lower than the $AF$ of SD. In general, the $AF$ data from PINC are at the lower end of the $AF$ range presented in Figure 5 and could result from

instrument differences discussed below. Other apparent differences in $AF$ between the SD and SD-SOA particles are discussed only after consideration of particle size in the discussion below. The uncertainties in $AF$ (barely visible on the log plot therefore not shown) are between 10-28% (see caption Figure 4 and methods) derived from aerosol particle and ice crystal counting statistics. This would imply that a difference in $AF$ by a factor of 2 or more is much greater than the maximum uncertainty in $AF$ (28%) and therefore significant. Given that poly-disperse particles were sampled in this work and that surface area plays

an important role in ice nucleation, we normalise the $AF$ for surface area to determine $INAS$ densities and reduce potential size biases (if any) that may cause the large spread in $AF$ for experiment 11 between the coated and uncoated dusts (see section 3.2). The number of large particles sampled by each chamber will influence the reported $AF$ since INP activity scales with dust particles size (Kanji and Abbatt, 2010; Lüönd et al., 2010)  thus large particles would contribute to the active INP population resulting in higher $AF$. As such it is not surprising that AIDA shows the highest $AF$ compared to PINC and CSU-CFDC because

of particle impactors applied upstream of PINC and CSU-CFDC which limit the largest particle sizes to 0.7 and 1.0 µm volume equivalent diameters, respectively. It is therefore also not surprising that the CSU-CFDC shows a somewhat higher $AF$ than PINC.

Time can also play an important role in promoting ice nucleation for both the deposition regime (Kanji and Abbatt, 2009) and immersion mode (Welti et al., 2012; Wex et al., 2014). The residence time in AIDA is the highest, on the order of a few minutes, whereas in PINC and CSU-CFDC it is much lower, on the order of 4-5 seconds. It could therefore be expected that the $AF$, which does not account for time, would be lower in PINC and CFDC compared to AIDA. It is not expected that the OPC cut-off sizes for ice crystal threshold size (see section 3.2 and 3.3) would influence the $AF$ for the $T$ and $RH$ conditions

reported here as growth rates to a few microns in diameter after nucleation of ice are expected to be almost instantaneous for $RH_w$ =105%.

## 3.2. INAS density of uncoated and coated Asian and Saharan Dust

The comparison using $INAS$ densities for poly-disperse aerosol samples is important to account for particle size because larger particles should be more ice active (i.e. can nucleate ice at lower $RH$ or supercooling (Archuleta et al., 2005; Kanji and Abbatt,

2010; Lüönd et al., 2010; Burkert-Kohn et al., 2017). The increase in $INAS$ density as a function of decrease in temperature for AD and SD are shown in Figures 6, 7 and 8. Compared to the $AF$ data where the maximum scatter of about a factor of 1000 is observed between 245 and 250 K, by accounting for the surface area, the scatter reduces to within a factor of 100 (factor of 10 considering uncertainties) in the $INAS$ density spectra. It is also evident from the $INAS$ results in Figures 7 and 8 that there is no systematic response of the ice nucleation activity of the dust particles to the presence of coatings, nor the

thickness (see Figure 6). An absence of the effect of organic (levoglucosan) and inorganic coatings on ice nucleation of mineral dust particles in the immersion mode has been reported previously (Sullivan et al., 2010a; Tobo et al., 2012; Kulkarni et al., 2014; Kulkarni et al., 2015) and discussed above (see section 1). There a few indications of the SOA coating impeding ice nucleation by reducing the $INAS$ of the coated particles, for experiments conducted with the CSU-CFDC and PINC. In particular, for $T > 240$ K, AD-SOA shows a lower $INAS$ density for both instruments by up to a factor of 10. At $T < 240$ K,

AD-SOA $INAS$ is lower than that of AD for PINC measurements, however, by a smaller margin. For both PINC and CSU-CFDC the differences are all within the $INAS$ uncertainties, and thus are not significant (overlapping error bars). The thickness of the coating for the ranges tested here do not appear to play a role either as shown in Figure 6. At ~238 K the PINC experiments 4 and 5 (Figure 7) have about the same $INAS$ density but the particles coating thickness is estimated to be 3 and 60 nm respectively. Similarly, at ~254 K the CSU-CFDC experiments 5 and 6 have coating thicknesses of 60 and 26 nm

respectively and $INAS$ for AD-SOA in experiment 5 is higher ($1.4 \times 10^9$ m$^{-2}$) than experiment 6 ($2.9 \times 10^8$ m$^{-2}$) suggesting that a thicker coating does not impede ice nucleation at these conditions.

The $INAS$ density for SD and SD-SOA in Figure 8 show increased scatter compared to the AD results, especially at temperatures of 238 and 243 K. At warmer temperatures, the scatter in $INAS$ density is largely reduced indicated by the

overlapping data and error bars. At the colder temperatures, PINC data are biased towards lower $INAS$ densities than CSU-CFDC. When considering the data obtained by each instrument separately, there is no difference between the ice nucleation ability of SD and SD-SOA, a consistent conclusion for all three instruments. Similar to AD, the coating thickness did not influence the ice nucleation ability of the SD particles as shown in Figure 6. Studies that reported partial (Sullivan et al., 2010b; Tobo et al., 2012) and complete (Augustin-Bauditz et al., 2014) deactivation in the immersion mode freezing of dust particles due to coatings of $H_2SO_4$ looked at coating thicknesses between $1 – 15$ nm, overalapping with the range of $3 – 60$ nm tested in this work. In particular, Augustin-Bauditz et al. (2014) report an influence on immersion mode ice nucleation activity of feldspar with the thicker $H_2SO_4$ coatings (15 nm) resulting in lower frozen fractions than those with thinner $H_2SO_4$ coatings (3 nm). Similar to conclusions drawn earlier by Sullivan et al. (2010b) who also found significant lowering of the $AF$ in the immersion mode when $H_2SO_4$ coating thickness increased progressively from 1, 2.4 to 4.1 nm with the $AF$ for the latter coating thickness being at the edge of the limit of quantification. On the other hand, studies that did not observe an effect of coating also do not report an effect of coating thickness. For example Kulkarni et al. (2014) report no effect of 1 and 40 nm thick $H_2SO_4$ coating on a variety of mineral dust particles in the immersion mode. We note that the coatings discussed in the studies above are produced by transiting particles through heated vapour regions of the coating substance. As such thicker coatings are associated with higher processing temperature which could have an effect on the ice nucleation activity by enhancing reactions between the coating and the substrate particles. Sullivan et al. (2010b), showed that thermodenuding $H_2SO_4$ coated ATD particles at 523 K further reduces their ice nucleation activity compared to particles with the same coating thickness without thermodenuding.

In our work, during sample generation, we cannot exclude the possibility of uneven coatings on the natural dust particles due to their highly irregular shape, size and porous features thus having potential for exposed mineral dust surface sites even in the coated particles. Assuming thicker coatings block water access to the dust surface, or require larger diffusion time scales of water through the coating, we would expect a suppression in $INAS$ density for thicker coatings compared to dust with thinner coatings, in particular over the short time scales in our flow chambers. However, this is not observed and could be due to the high supersaturated $RH_w$ used to evaluate the maximum ice nucleation in the immersion mode. The SOA coatings were formed and condensed onto the AD and SD at room temperature and at very dry conditions of $RH_w < 10$ % implying that the coatings could have formed viscous (Maclean et al., 2017) glassy phases which have been reported to readily form at $\sim 298$ K and $RH_w << 50$ % (Wagner et al., 2012). In particular, Kidd et al. (2014) have shown specifically for $O_3/\alpha$-pinene SOA that if formed at low $RH$ as was done in our work, the SOA remains in a highly viscous and solid state even upon significant increase of $RH_w$ up to 85-90%. The transition $RH_w$ to liquid SOA being higher than reported elsewhere (Liu et al., 2016; Ye et al., 2016; Gorkowski et al., 2017; McFiggans, 2018) likely due to the low temperature during exposure to high $RH$ in our ice nucleation experiments. Indeed the transition from solid to a semi-solid/glassy state for a variety of organic compounds forming internally mixed SOA as would be expected with $O_3/\alpha$-pinene SOA (Hallquist et al., 2009), is reported to occur at $RH_w = 75-85$ % over the temperature range $250 – 230$ K (Wang et al., 2012a). For the same temperature range, other estimates of SOA proxies find

the transitions to liquid from a glassy state occur at $RH_w < 85\%$ (Ladino et al., 2014) and 50-65 % (Koop et al., 2011). Given that we report data for conditions of $RH_w \geq 100\%$, water condensing on the particles likely penetrates or dissolves the glassy state SOA by diffusing into the SOA coating thus forming dilute droplets which freeze heterogeneously via the immersion mode. The hygroscopicity parameter, $\kappa$, of $\alpha$-pinene SOA has been reported to be 0.10-0.14 (Jurányi et al., 2009) compared to

0.017 (Augustin-Bauditz et al., 2016) for uncoated dust particles. Therefore, oxidised organic coatings such as the SOA used here can even enhance water uptake of the coated dust particles thus favouring immersion mode freezing as opposed to suppressing ice nucleation in the deposition mode (Möhler et al., 2008). The presumed highly viscous nature of the SOA formed at the conditions used here could leave parts of the mineral surface exposed and thus able to act as an INP. The possibility of incomplete coatings in our sample generation could explain the similarities in *AF* or *INAS* density of the coated

and uncoated dust. This would explain why a difference in *INAS* density is not consistently observed for coated dusts compared to uncoated dusts. Lastly, it has been observed that glassy naphthalene SOA served as immersion INPs after water uptake for 243 K > *T* > 233 K and at water saturation conditions (Wang et al., 2012b). For the same particles at *T* > 243 K however, water uptake occurred between $RH_w$ of 80-90% depending on the O/C ratio but did not freeze upon reaching water saturation. The authors concluded from water diffusivity calculations that at the warmer temperatures water penetrates and dissolves the SOA

that would otherwise act as an immersion INP at the colder temperatures (Wang et al., 2012b). In this work, solvation of the SOA coating can still render the particles ice nucleation active due to the mineral dust core, as indicated by the observed *AF* and *INAS* density for temperatures as warm as 253 K.

Experiments with similar SOA coatings as used in this work were reported for illite clay mineral particles and ATD from the

AIDA chamber (Möhler et al., 2008) for a lower temperature range of $210 - 215$ K. It was found that for the same $RH_i = 120\%$, the SOA coated ATD yielded an *AF* of a factor of 15 lower than uncoated ATD. For illite particles, almost complete activation ($AF \sim 1$) was observed at $RH_i = 120\%$, but the SOA-coated illite did not show any ice formation until $RH_i = 160\%$ was reached (homogeneous freezing conditions at the temperature reported), suggesting that under these conditions ($RH_w << 100\%$, deposition mode regime) the organic coating significantly suppresses ice nucleation for dust particles (Möhler et al., 2008).

Considering the temperature and *RH* range (deposition mode regime) of the study, adsorbed water may not be able to diffuse through and solvate the SOA coatings, on the short observation time scales of AIDA (~120 seconds during an expansion), suggesting that the ice nucleating active sites on the surface of the dust are blocked from potentially stabilizing ice germ formation. Furthermore, the difference in magnitude of ice nucleation suppression could be attributed to the fact that ATD likely had much thinner coating (mass fraction of SOA to ATD 17 %) compared to a thicker coating on illite with a mass

fraction of SOA of 41% (Möhler et al., 2008). The absence of a SOA coating effect on the immersion freezing of dust particles observed here could imply first, the organic coating produced here did not modify the active sites on the mineral dust surface thus re-exposing them upon dissolution into the droplet, and second, if the surface was modified, equally potent active sites were exposed on the mineral dust surface, retaining its avergae ice nucleation activity. This is in contrast to the effect of

coatings with $H_2SO_4$ that were shown to irreversibly modify mineral dust INPs in the immersion mode (Sullivan et al., 2010b; Tobo et al., 2012; Augustin-Bauditz et al., 2014).

### 3.3. Comparison of coated to uncoated dusts for the three INP chambers

During the measurements conducted in this work, one of the goals was to achieve an inter-comparison of sampling INPs between the techniques used. To do this, in Figures 9 and 10 we have plotted data for the *INAS* density of coated versus uncoated dusts without distinguishing between AD and SD, for each of the three instruments used. Here, the influence of the method and instrument type on any observed differences in ice nucleation activity between the coated and uncoated particles should be normalised. To match the coated to the uncoated *INAS* densities, the data are binned into 1 K bins ($\pm$ 0.5 K) for each instrument used (Figure 9). To show the temperature regime where the difference in *INAS* density between coated and uncoated dusts is the highest, the data were binned only by temperature but without distinguishing for the instruments (Figure 10). We note that based on the earlier discussed comparisons (see end of Introduction and end of Section 3.0), the CSU-CFDC and PINC data have been increased by a factor of 3, to account for the low bias of INP concentrations and *AF* that is typical in CFDC INP counters as discussed by DeMott et al. (2015).

Figure 9 reveals that for $INAS_{(SOA,\,coated)} > 10^{10}$ m$^{-2}$ (*y*-axis), all data points (except one outlier) lay onto or overlap with the 1:1 line considering uncertainties (shown as the error bars) suggesting no differences in ice nucleation activity between coated and uncoated dusts irrespective of instrument used or dust type sampled. Below this value, fewer data points lay on the 1:1 line with a visible bias to lower *INAS density* regardless of the instrument used in this study suggesting a slight bias towards coated dusts having an impeded ice nucleation ability in the range $INAS_{(SOA,\,coated)} < 10^{10}$ m$^{-2}$. Furthermore, the ice nucleation suppression appears to be higher for CSU-CFDC and PINC, than for AIDA with the latter having all data points within a factor of 10 of the 1:1 line, but a few data points being outside this factor of 10 region for PINC (1 for CSU-CFDC). The difference between the continuous flow chambers and AIDA may be attributed to the shorter residence times of 4-5 seconds compared to that of minutes in AIDA, which would allow diffusion of water through and possibly dissolution of the SOA coating for immersion freezing to occur in this temperature range. The fraction of large particles (> 1 μm) sampled by AIDA was also higher and could contribute to the observed higher *INAS* of the coated dusts. Compared to AIDA and CSU-CFDC, PINC *INAS* density shows positive bias towards uncoated dusts. In addition, two data points, one from each PINC and CSU-CFDC, lie outside the factor 10 line (Figure 9) even considering uncertainties, which may suggest that the coatings on the dust may occasionally impact the immersion freezing activity (for example for a complete thick SOA coating). However, given the large spread of data in the range $10^8 < INAS_{(SOA,\,coated)} < 10^{10}$ m$^{-2}$, it is not possible to disentangle beyond the discussion in sections 4.1 and 4.2 as to what the reasons could be, for the sometimes observed suppression in *INAS* density of the coated dusts. From a previous comprehensive intercomparison study on the immersion freezing of illite particles (Hiranuma et al., 2015) agreement between 17 ice nucleation instruments was found within a factor of 10 in *INAS* density. Given 95 % of the data points in Figure 9 lay within or overlap (with uncertainties) the factor 10 region of the 1:1 line suggests that there is no

difference in ice nucleation ability of the coated and uncoated dusts and any small differences observed could be instrument or experiment specific.

Furthermore, the coated versus uncoated dust *INAS* densities plotted in Figure 10 do not distinguish for the instrument used but show the temperature dependency instead. Similar to the previous discussion, a higher spread in the data from the 1:1 line is observed for warmer temperatures (lower *INAS* values). A factor contributing here again is larger particles with varying composition can bias towards higher *INAS* values. We note that there is also some spread at $INAS < 10^{11}$ m$^{-2}$ suggesting that uncoated dust particles have higher *INAS* values. In addition, 80% of the data fall within the factor of 10 of the 1:1 line (Figure 10). Fitting the data with a linear fit (with a forced (0,0) intercept) yields a slope of $< 1$ suggesting a slight bias towards the ice nucleation activity of uncoated dusts being higher than coated dusts. However, the 66% (95%) confidence interval prediction bands of the fit show that 75% (97%) of the data are predicted by this fit suggesting that overall there is no influence on the immersion mode ice nucleation activity caused by the SOA coatings used in this study for both types of desert dust.

### 3.4. Comparison of ice nucleation properties of Asian and Saharan dusts

In Figure 11, we show the *INAS* for AD and SD as a function of temperature. We note there is no significant difference between the two dusts at the $p < 0.05$ level ($t$-value = 1.24, $p$-value = 0.11) in their ice nucleation activity at these conditions. This is especially visible in Figure 11 considering experimental uncertainties as shown by the error bars that mostly arise from particle counting and sizing. Fitted curves to the AD and SD data separately resulted in completely overlapping confidence interval bands at the 95% level (not shown here), supporting the lack of difference in ice nucleation activity between AD and SD. PINC sampled the smallest size fractions of the three chambers since the volume equivalent cut-off diameter for its impactor was 0.7 μm (1.0 μm for CSU-CFDC). Whereas AIDA did not sample downstream of an impactor and thus a substantial amount of supermicron particles were sampled. These size cut-offs an result in the lower ice nucleation activities in general observed in PINC, given the ice nucleation activity increases with particles size for a given dust sample type (Archuleta et al., 2005; Welti et al., 2009).

At the warmer end of the temperature range ~253 K, it appears that SD has lower *INAS* than AD, however, there is some scatter in the data and also significant overlap between the uncertainties of the data points. One case where AD *INAS* density is much higher than the rest is for experiment 2 with CSU-CFDC where a lower impactor cut-off size was used (0.7 μm) for this particular experiment before the impactor cut-off was adjusted to the 1.0 μm cut-off size stated above. This would matter for *INAS* density if the particle composition changes with size, which can be the case with mineral dust samples (Vlasenko et al., 2005). However, harder minerals such as feldspars and quartz tend to be enriched in the larger size fractions of the samples and are also the more ice active particles (compared to softer clay particles), thus the composition change with size would not explain the high *INAS* density at this temperature for this experiment. In general from the mineralogy, both AD and SD contain clays, feldspars and quartz (Linke et al., 2006; Boose et al., 2016b), with the AD containing significantly more $SiO_2$ (70.4 %,

Möhler et al., 2006) than SD (26.8 % Möhler et al., 2006). From the mineralogy, we infer that the AD sample contained more feldspars (25 % by mass, Boose et al., 2016b) than the SD sample where Linke et al. (2006) report qualitatively (without mass percentages) that there is a significantly higher fraction of dolomite ($CaMg(CO_3)_2$), calcite ($CaCO_3$) and gypsum ($CaSO_4$) compared to feldspars. This is consistent with the overall mineralogy reported in Boose et al. (2016b), for Saharan dusts

sampled in Morocco, Egypt and Tenerife, that there is less feldspar fraction by mass compared to clays, calcites and muscovite in SD compared to AD. Given the general overlap in data across the temperature range sampled and the instruments used, a difference in ice nucleation activity between the AD and SD samples investigated here is not supported. In Figure 11 we also fit the data with an exponential function to predict the *INAS* density as function of temperature with the following:

$$INAS_{density} = e^{a+bT} \qquad (1),$$

where *INAS density* is in active sites per $m^2$ and valid for $255 < T < 232$ K, with fit coefficients of $a = 121.4$, $b = -0.403$. This is compared to the parameterization for desert dusts proposed by Niemand et al. (2012, N12) also shown in Figure 11 for which $a = 8.934$, $b = -0.517$ and with $T$ in °C. Compared to N12, at warmer temperatures we slightly overestimate *INAS* density by

a factor of 2.5 and at the colder temperatures the fit underestimates *INAS* density up to a factor of 4.5 compared to N12. The slope of the fit for this work is slightly lower than N12, implying a slightly lower temperature dependency than predicted by N12. However, given the overall scatter in the data for the three chambers used and the uncertainties associated with *INAS* density, there is good agreement between the fit proposed here and that of N12.

## 4. Conclusions

We present the ice nucleation ability in the immersion mode of two types of natural desert dust samples that were uncoated and coated with the dark ozonolysis products of $\alpha$-pinene as a proxy for atmospheric secondary organic aerosol (SOA). We conclude that the SOA coatings did not affect the immersion ice nucleation ability of the dust particles in the temperature range 253 to 235 K irrespective of coating thickness (3 – 60 nm). This suggests that coatings forming on atmospheric airborne dust during transport via condensation of SOA resulting in ageing will not significantly change the immersion freezing ice

nucleation properties of dust particles. Furthermore, the thickness of the coatings in the studies presented here cover a wide range (see Figure 6) and suggest that in the atmosphere even appreciable amounts (tens of nm) of organic coatings (e.g., acids) on dust particles (Sullivan and Prather, 2007) should not impede immersion freezing. We note that we only used two types of mineral dust samples and one proxy of SOA for coating, but given the results from this work and those discussed above from previous work, we cannot assume that condensation of coatings or other chemical processing will impair immersion mode ice

nucleation properties of mineral dust particles. Immersion mode ice nucleation is less sensitive (if at all) to coatings and ageing based on this and prior studies on mineral dust. These results are similar to the INP properties in the immersion mode for dust

particles exposed to $HNO_3$ or in some cases $H_2SO_4$ vapours that also showed lack of an impairment in immersion freezing ability. As the coated particles are activated into droplets prior to experiencing freezing in the ice nucleation chambers, the coatings may have dissolved to reveal the particle ice active surface sites. Observations of scatter between the three INP chambers, PINC, CSU – CFDC and AIDA can be attributed to differences in the evaluation of immersion freezing in

continuous flow diffusion chambers where *AF* (by extension *INAS* density*)* can for some INPs be a factor of 3-9 lower due to aerosol spread outside of the focussed sample lamina (DeMott et al., 2015; Garimella et al., 2017) than for expansion chambers like AIDA. Considering the time scale of seconds in the CFDC type instruments compared to minutes in AIDA, and the particle size range sampled due to the use of impactors, the results obtained between the INP chambers can be considered to be within good agreement (overlapping error bars) for the *T* and *RH* conditions evaluated here. To circumvent confounding effects of

particle size and coating thickness, future studies with size-selected particles and carefully controlled coating thickness would be desirable to exclude uncertainties in how particle size and composition influence the ice nucleation activity. Furthermore, *INAS* spectra of AD and SD obtained here were used to obtain a fit for the ice nucleation activities of the desert dusts. Differences in the ice nucleation activity of AD and SD were not considered significant here in light of the similar *INAS* density (within uncertainties and instrument scatter), similar to conclusions reached by DeMott et al. (2015) for the same dust samples.

Comparing the bulk (average) mineralogy to assess the particle mineralogy sampled by the cloud chambers which were mostly in the sub-micron to low super-micron size range imposes a limitation in assessing the effects of mineralogy on the ice nucleation activity given that single particles can have different specific compositions. Lastly, the assumption of a constant composition with size that is implicit in the use of *INAS* density to evaluate the results presented here may bias some of the compared results, but aids in the interpretation of poly-disperse samples where accounting for surface area is important.

*Author Contributions*: ZAK wrote the manuscript, synthesized the figures, conducted and analyzed PINC measurements. RCS, PJD and AJP conducted measurements on the CSU-CFDC. MN and OM conducted measurements on the AIDA chamber. HS prepared the coating experiments and provided coating thicknesses of dust particles. MN and OM provided size distribution data and hosted the entire measurement campaign at the AIDA facility. ZAK interpreted the ice nucleation data with

contributions from MN, PJD, OM and RCS.

*Competing Interests*: The authors declare no conflict of interest.

*Acknowledgements*: The authors would like to acknowledge Lothar Shütz from University of Mainz for the AD2 sample. ZAK

acknowledges the funding from the SNF grant number 200021_127275 and Ulrike Lohmann. C. Chou acknowledges funding from GAW-CH 2010 – 2013.  PJD, AJP, and RCS acknowledge support from U.S. National Science Foundation Grants ATM-0611936 and ATM-0841602, and travel support from the EUROCHAMP-2 Transnational Access proposal E2-2009-09-15-0009 (PI: Jonathan Crosier) funded within the EC 7th Framework Program. OM, MN and HS acknowledge funding from the

Helmholtz Association through the Virtual Institute on Aerosol-Cloud Interactions (VI-ACI, VH-VI-233). Support by the AIDA technical team for preparing and conduction the cloud chamber experiments is gratefully acknowledged.

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

**Table 1. Experiment number and corresponding SOA coating thickness calculated based on mass yield of SOA distributed equally over available dust surface area**

| Instrument and Experiment Number | Dust type – SOA coating thickness (nm) |
|---|---|
| CSU-CFDC/PINC | |
| 04 | AD – 3 |
| 05 | AD – 60 |
| 06 | AD – 26 |

|     |          |
| --- | -------- |
| 07  | AD – 9   |
| 08  | AD – 15  |
| 09  | SD – 12  |
| 11  | SD – 7   |
| 13  | SD – 8   |
| AIDA<br>07<br>09<br>11<br>39 | AD – 11<br>AD – 6<br>AD – 60<br>SD – 6 |

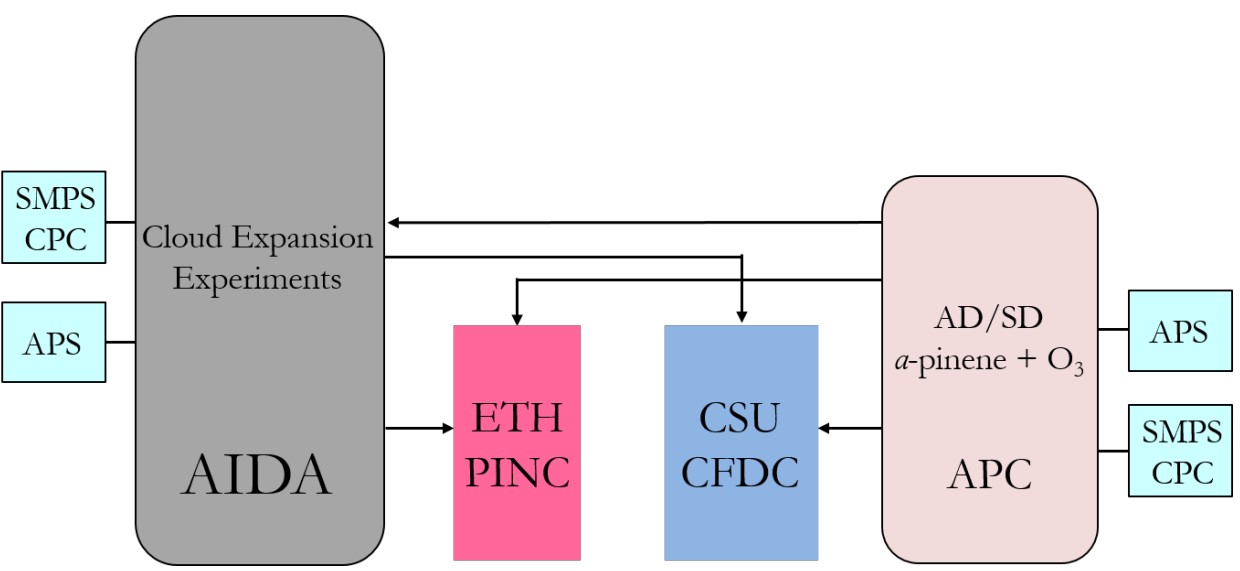

**Figure 1. Experimental schematic of INP and aerosol sampling and sizing by cloud chambers and counting instruments at the AIDA facility at the Karlsruhe Institute of Technology during the ACI-03 workshop in October 2009. APS: aerodynamic particle sizer, SMPS: scanning mobility particle sizer, CPC: condensation particle counter. See text for details on AIDA, ETH PINC and CSU-CFDC.**

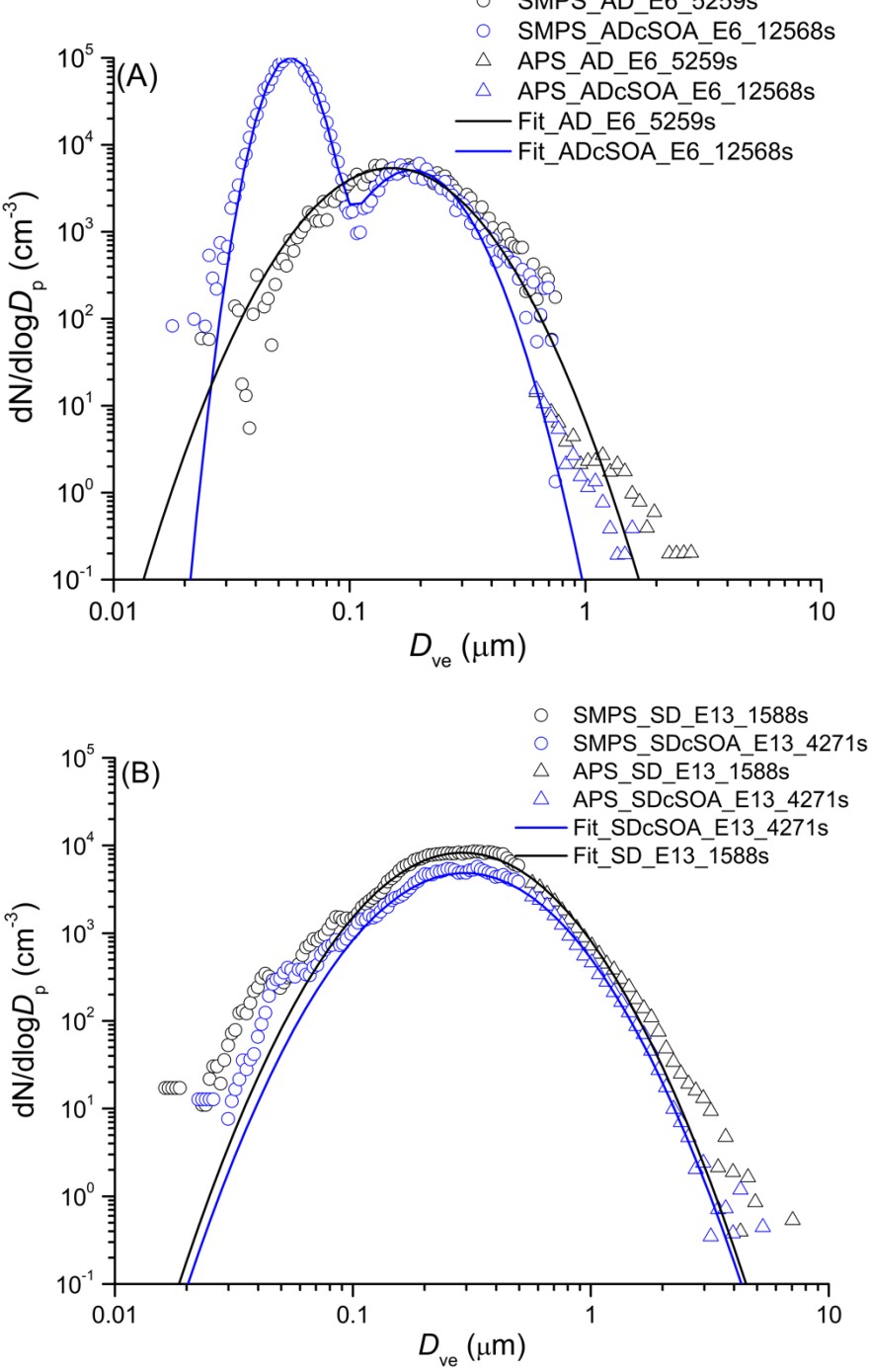

**Figure 2.** Exemplary number size (volume equivalent diameter) distribution of AD (panel a) and SD (panel b) sample before and after coating with α-pinene SOA. Nucleation of pure SOA particles results in the bi-modal distribution after SOA coating giving rise to the mode at 40 nm.

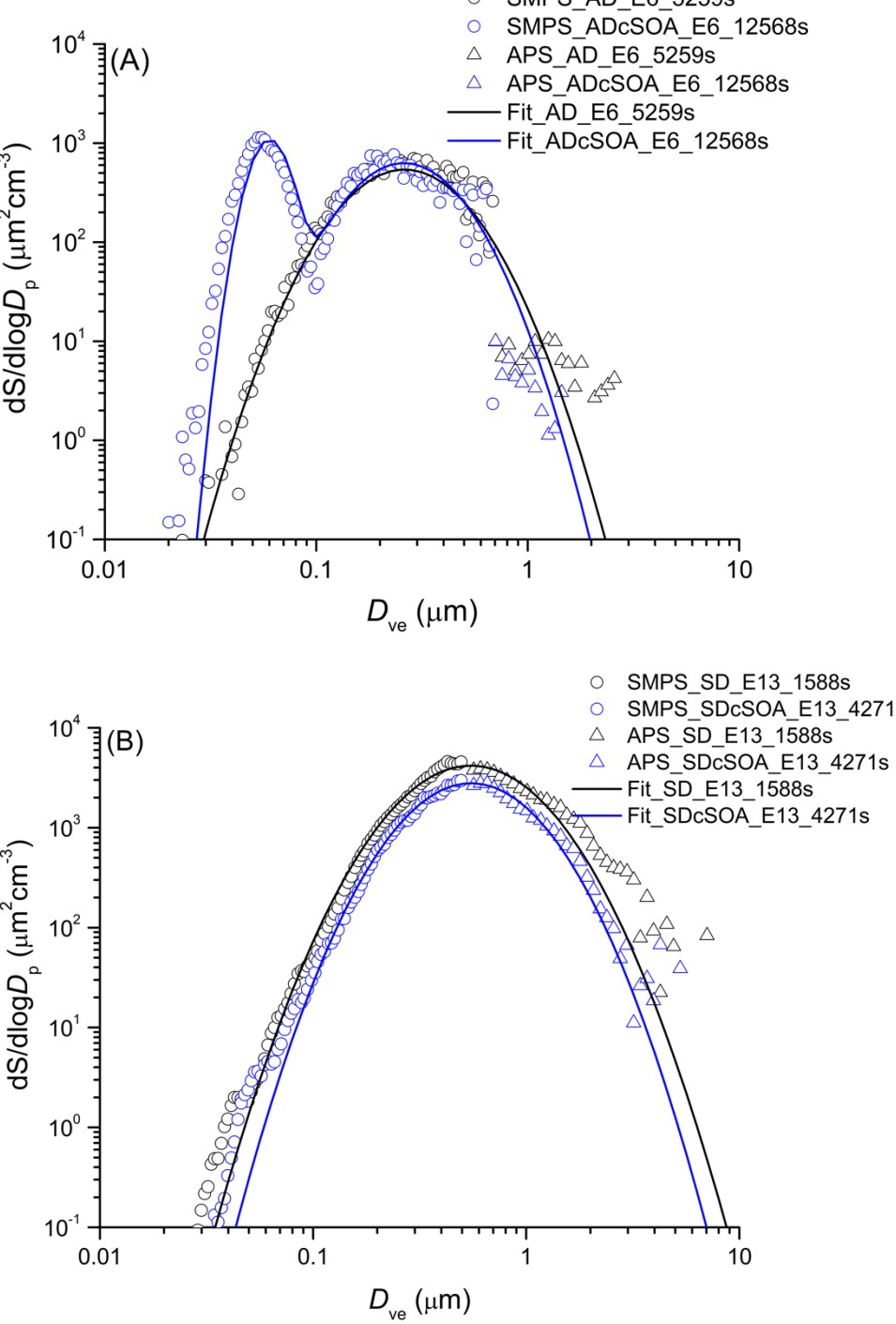

**Figure 3. Same as Figure 2 but for surface area distribution as a function of volume equivalent diameter.**

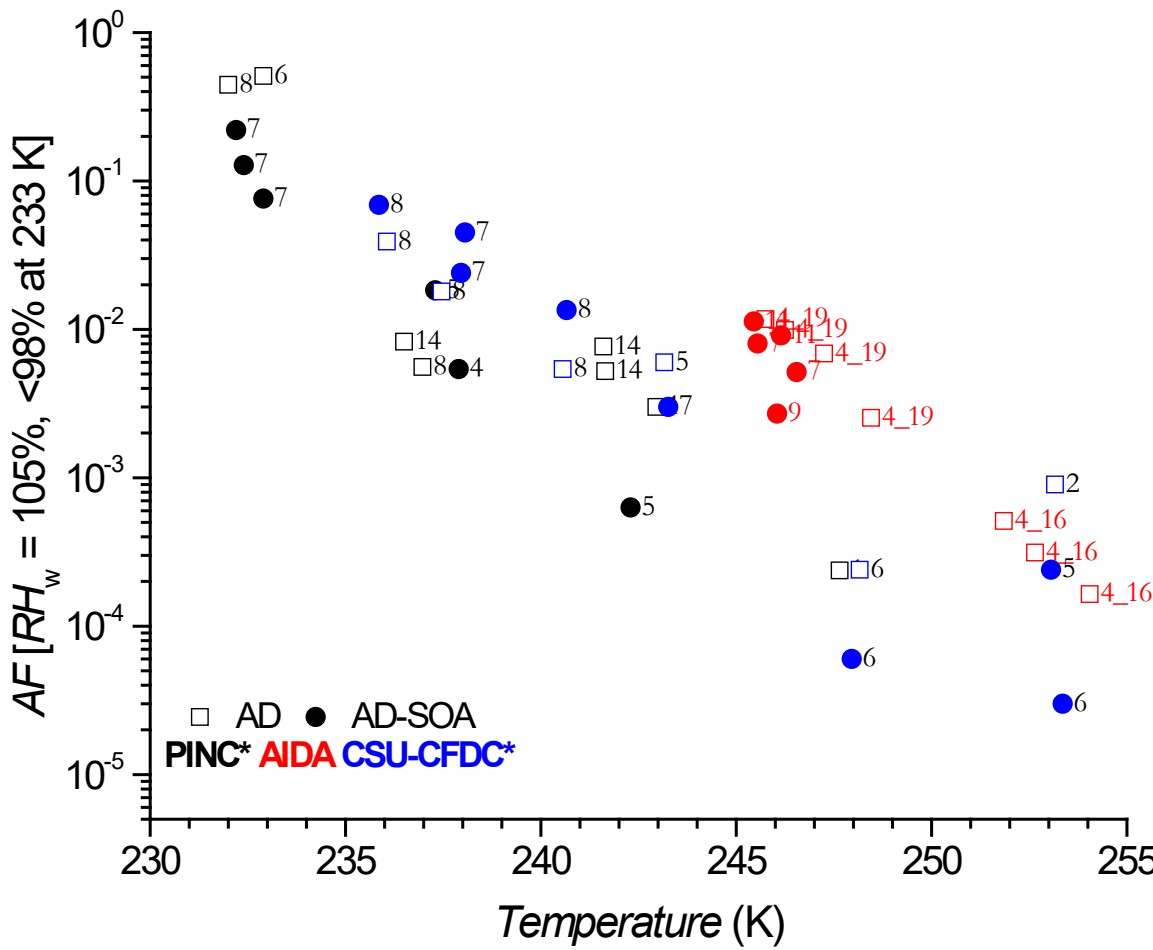

**Figure 4.** Activated fraction (*AF*) as a function of temperature for Asian Dust (AD, open squares) and AD coated with SOA (AD-SOA, filled circles). Numbers next to data points refer to experiment number. Coating thicknesses are shown in Table 1. *The PINC and CSU-CFDC data have been increased by a factor of 3 (see text section 4 for details). Error bars for *AF* of 10% (CSU-CFDC), 14% (PINC) and 28% (AIDA) when plotted are hardly visible (not shown).

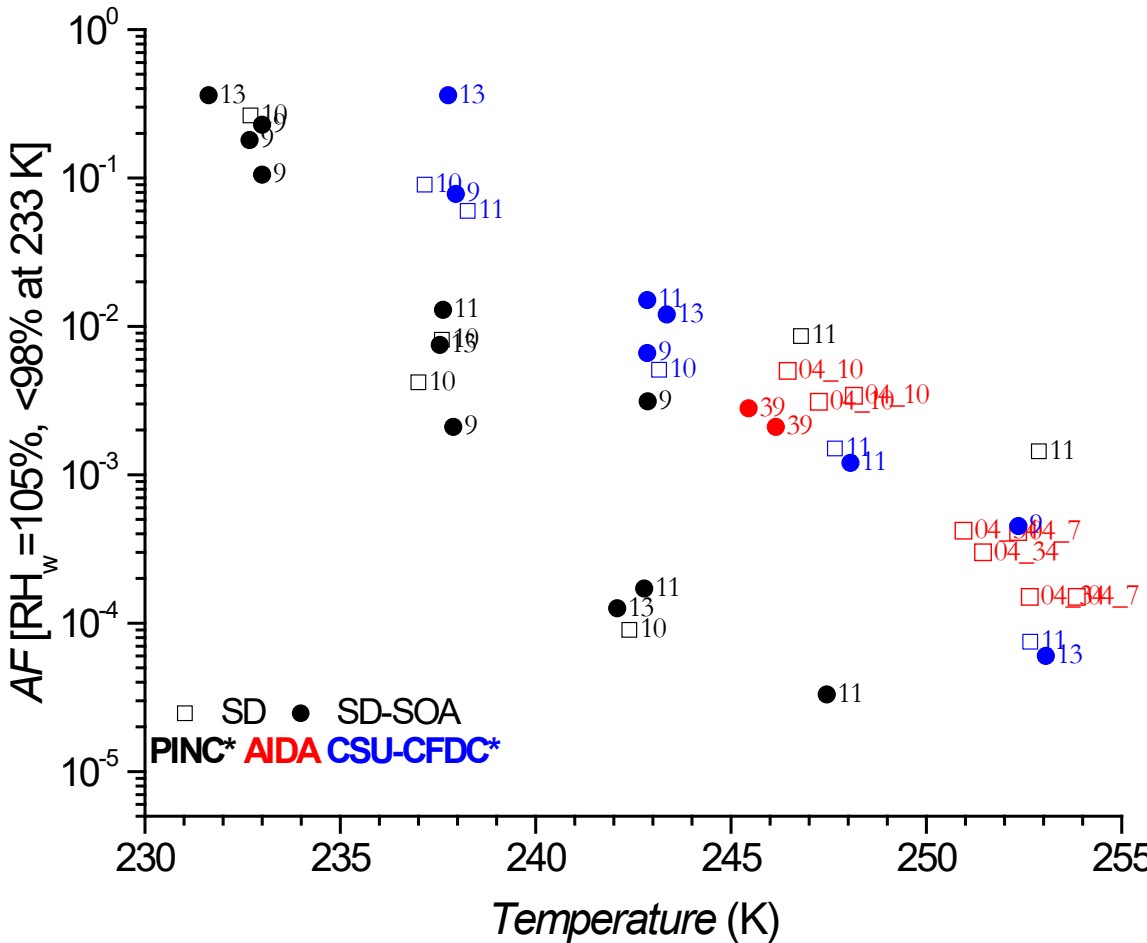

**Figure 5. Same as Figure 4, but for Saharan dust (SD) and SD coated with SOA (SD-SOA) particles.**

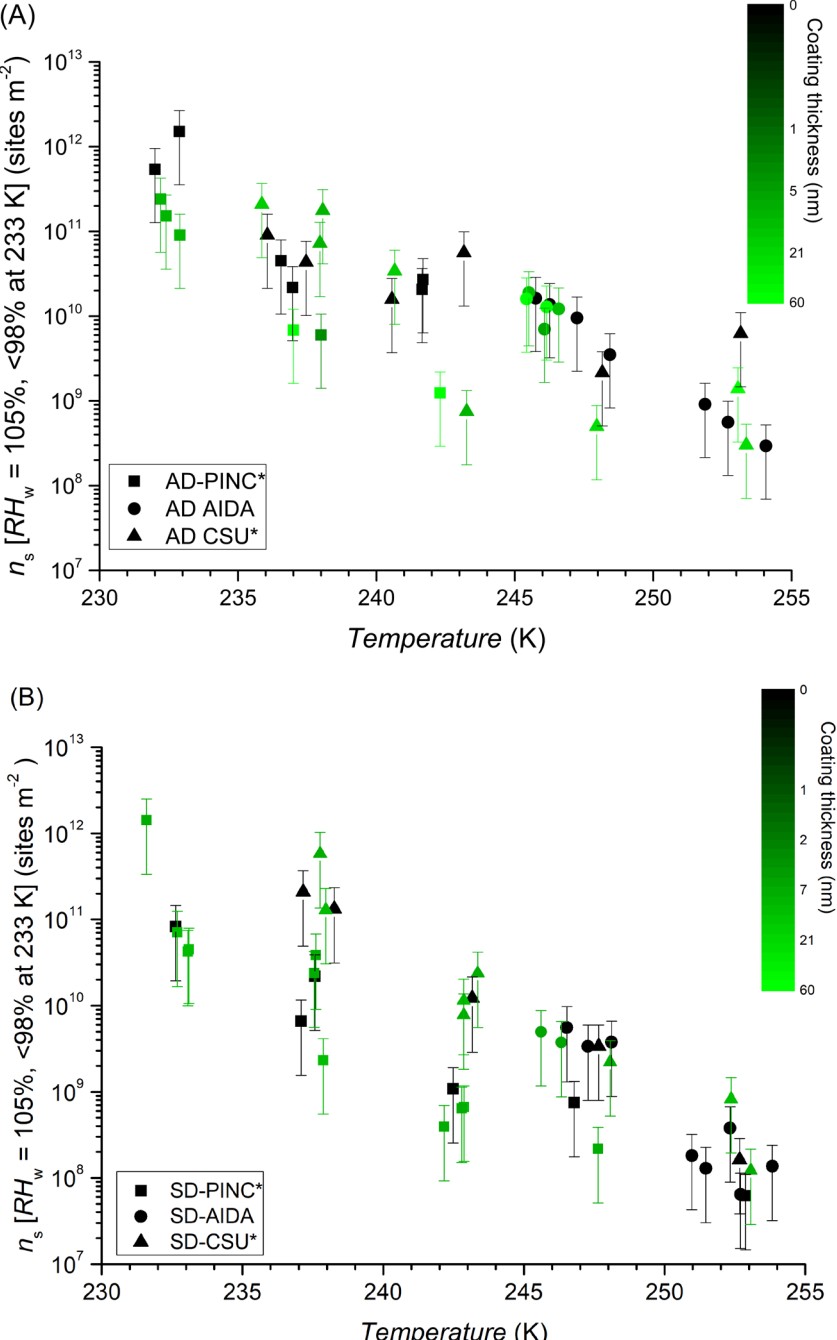

**Figure 6.** *INAS* **densities as a function of temperature and SOA coating thickness for AD (panel a) and SD (panel b). Black symbols represent uncoated dust. Error bars are based on 75% sizing due to spherical assumption and surface area determination and 15% from particle counting resulting in a relative error of 77% in** *INAS* **density. *The PINC and CSU-CFDC data have been adjusted by a factor of 3 (see text section 4 for details).**

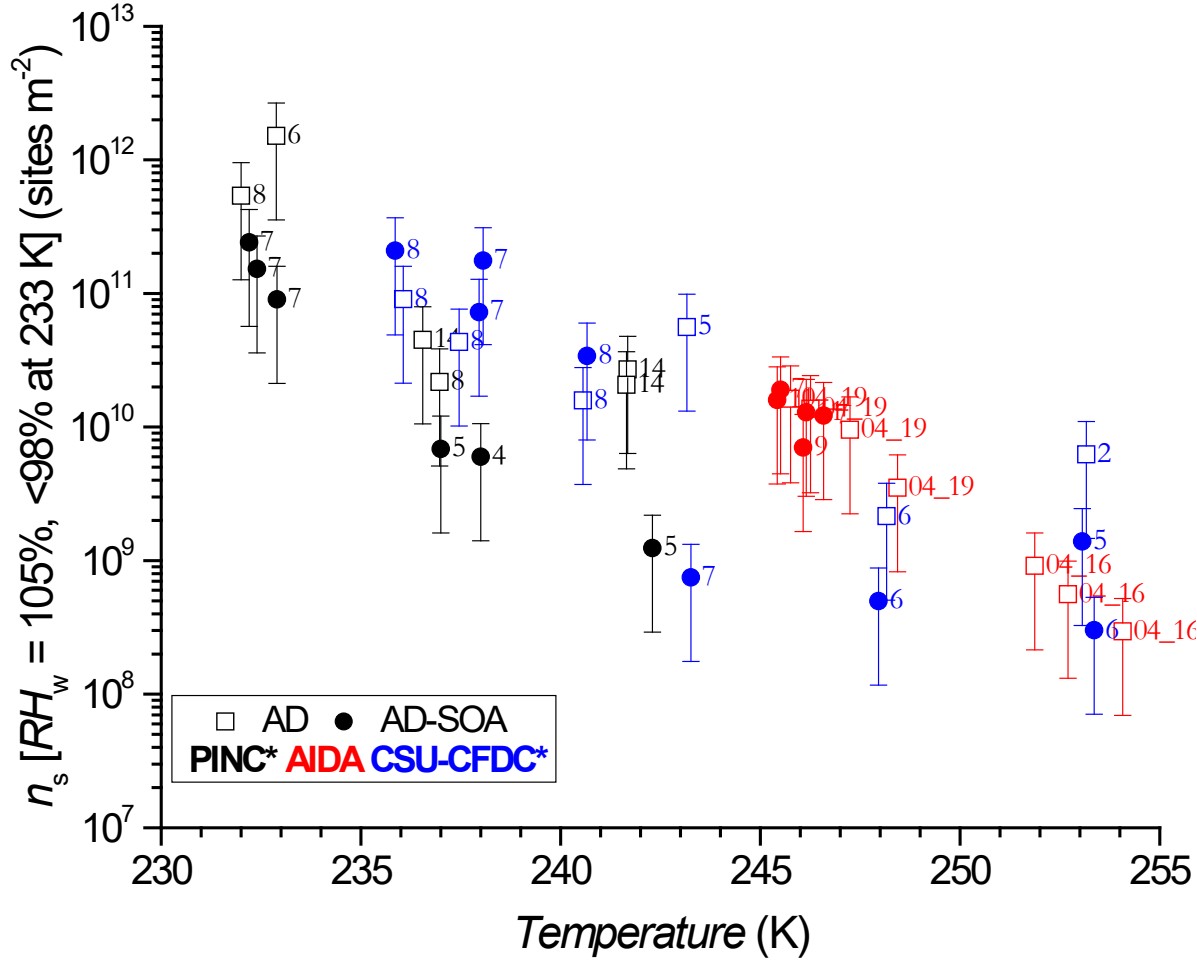

**Figure 7.** *INAS* densities as a function of temperature for AD and AD-SOA. Error bars are based on 75% uncertainty in particle sizing due to spherical assumption and surface area determination and 15% uncertainty from particle counting resulting in a relative error of 76% in *INAS* density Numbers next to data points refer to experiment number records for coating thicknesses shown in Table 1. *The PINC and CSU-CFDC data have been adjusted by a factor of 3 (see text section 4 for details).

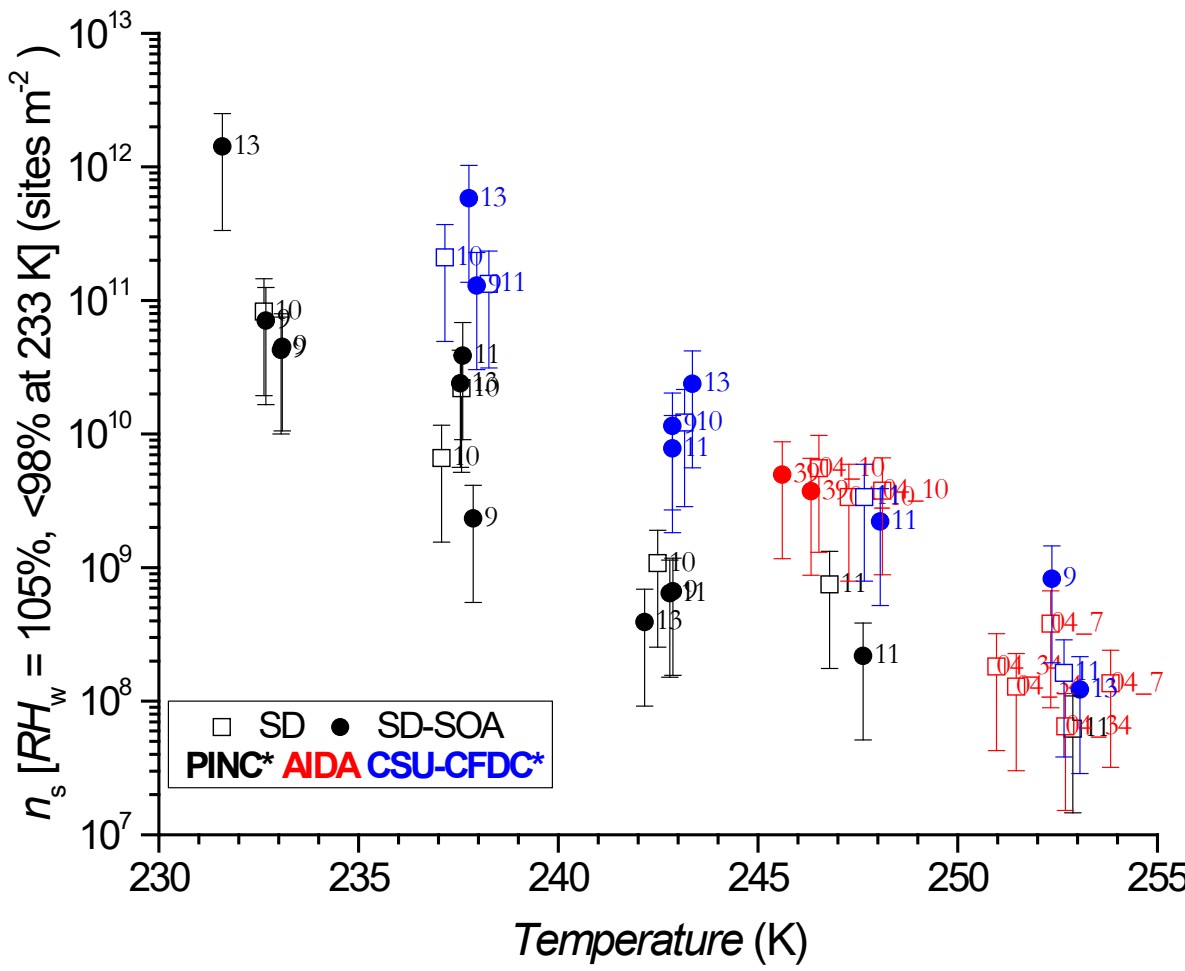

**Figure 8. Same as Figure 7 but for Saharan dust (SD) and SD coated with SOA (SD-SOA) particles.**

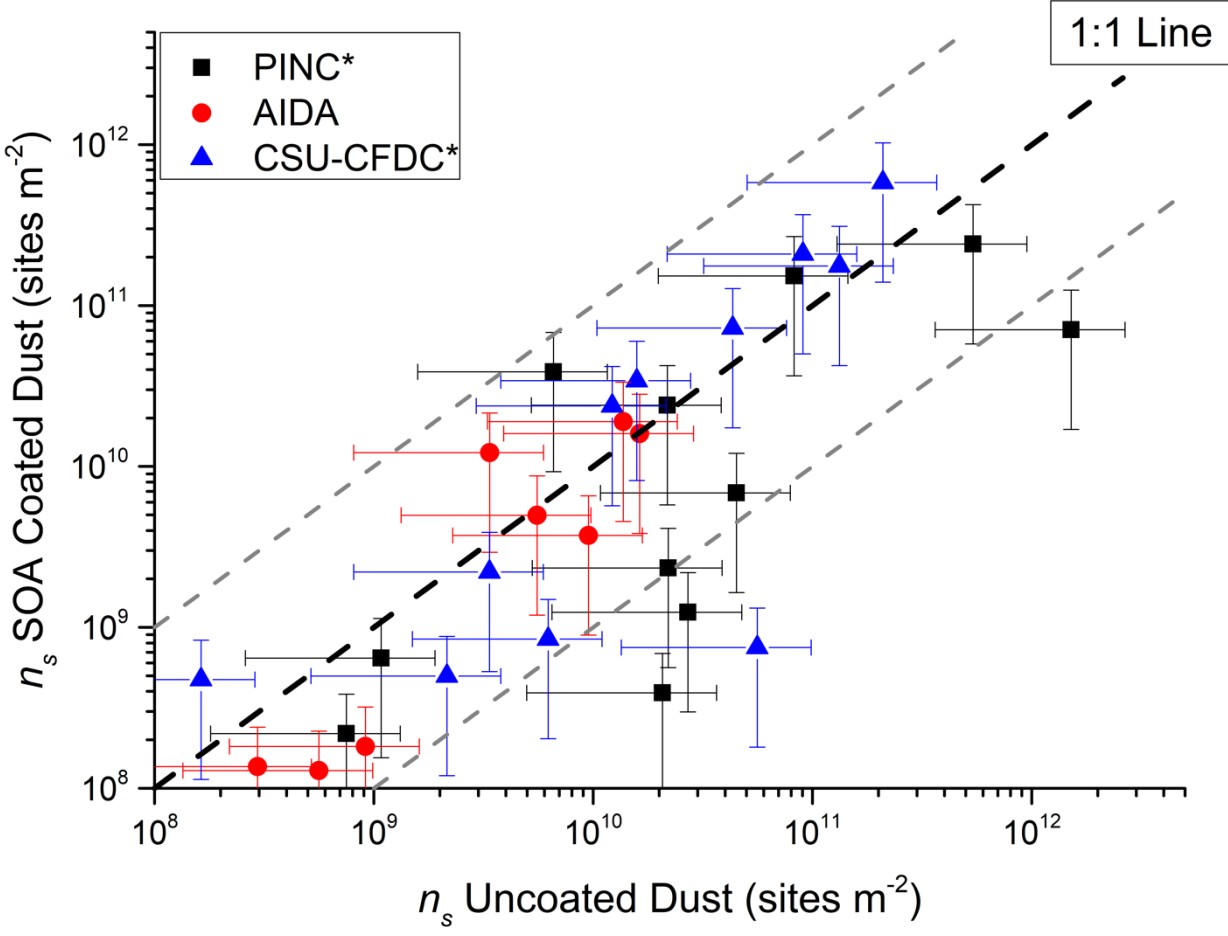

**Figure 9. Comparison of ice nucleation activity of uncoated dusts with coated dusts distinguished by the three cloud chambers used in this study. Data are binned by temperature into 1 K bins for each cloud chamber. The grey dashed lines represent a factor of 5 from the 1:1 line. Error bars in ns represent a relative uncertainty in sizing and counting particles of 76% (upper estimate).**

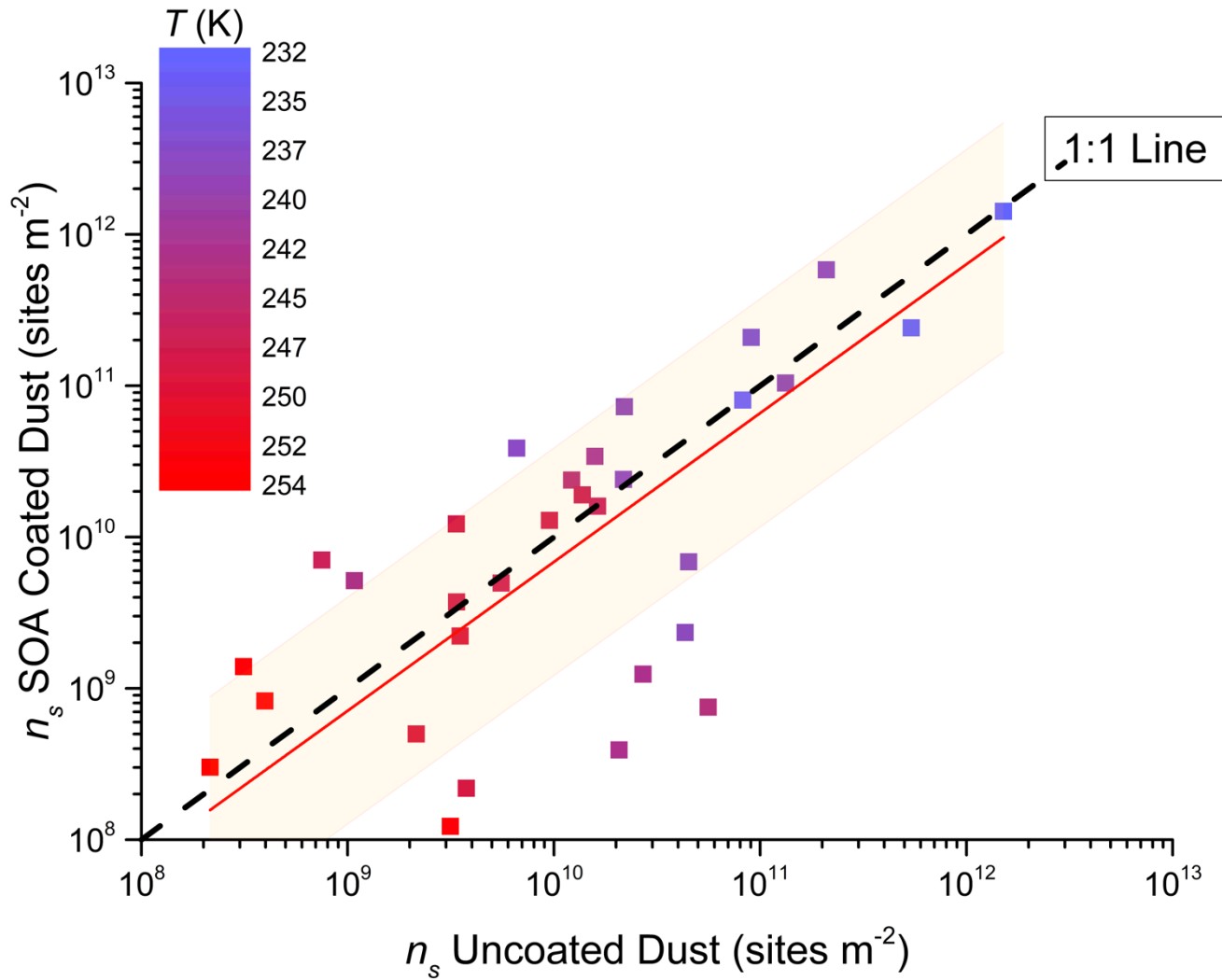

**Figure 10.** Same as Figure 8 but binned by temperature in to 1 K bins not accounting for cloud chamber type as a function of temperature. Error bars omitted for clarity of temperature color bar. Dashed line is 1:1 fit and red line is linear fit to data $n_s$(coated) $= n_s$(uncoated) with forced intercept (0,0) with $r^2 = 0.99$. Shaded region represent 66% prediction bands of fit.

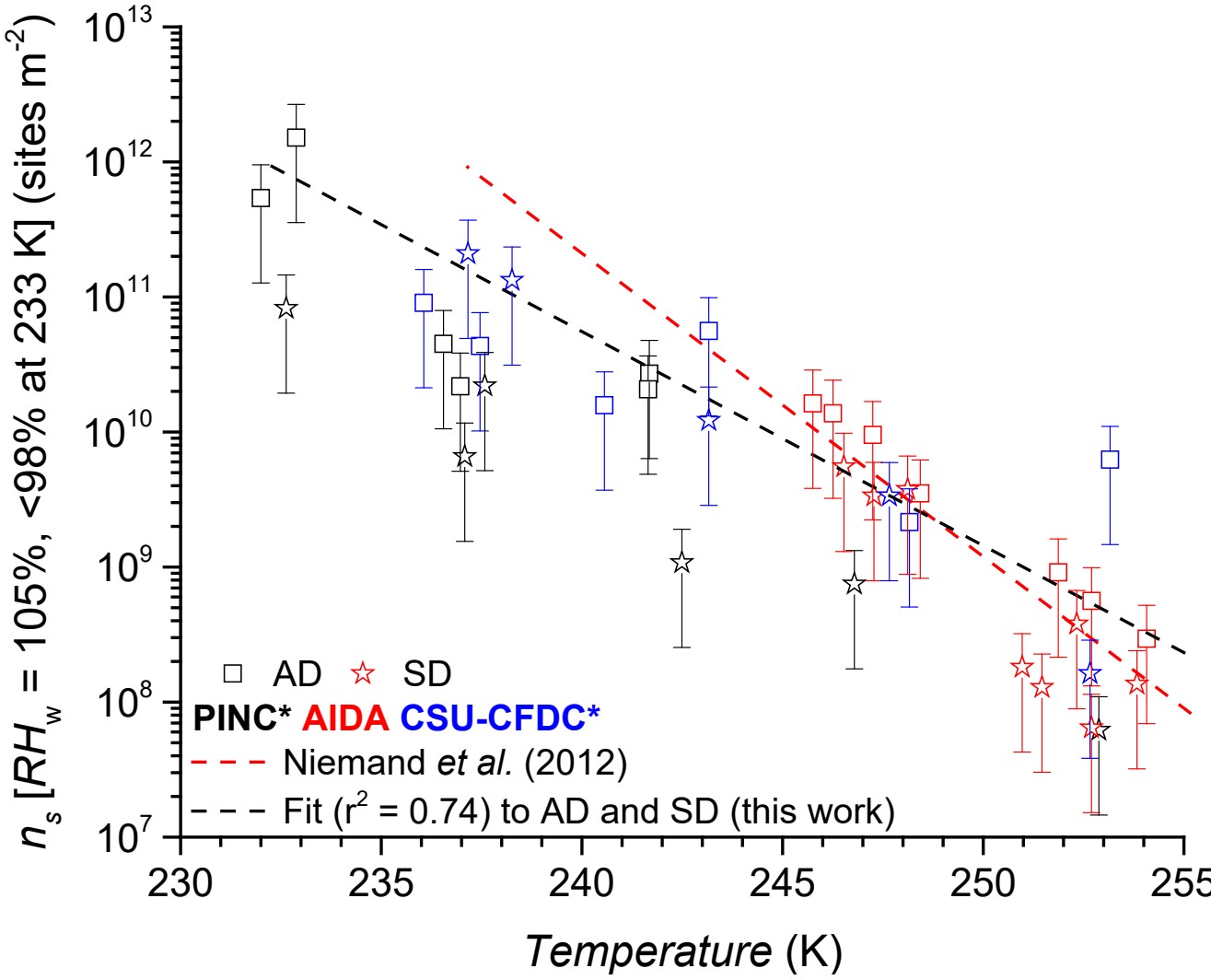

**Figure 11. INAS densities as a function of temperature for AD and SD. Error bars are based on 75% sizing due to spherical assumption and surface area determination and 15% from particle counting resulting in a relative error of 77% in ns.*The PINC and CSU-CFDC data have been adjusted by a factor of 3 (see text section 4 for details). Equation fore exponential fit (black dashed line) shown in text section 4.4.**