# Peer review of "Heterogeneous Ice Nucleation Properties of Natural Desert Dust Particles Coated with a Surrogate of Secondary Organic Aerosol"

_Atmospheric Chemistry and Physics, 2018_

## Referee Comment (RC1) · Anonymous Referee #1 · 17 Oct 2018

General comments:

This article reports that coating two types of mineral dust particles with a secondary organic aerosol proxy produced by dark ozonolysis of alpha-pinene made little difference to their immersion mode ice nucleating ability between 253 K and 233 K. No systematic differences in ice nucleating abilities of the dusts were observed with various atmospherically plausible coating thicknesses. Measurements were conducted simultaneously on three well established cloud chamber instruments with broadly consistent results. The study is relevant to the scope of ACP, presents useful and interesting new data and is scientifically sound. The paper is mostly well written. I have a few minor

comments, but am quite happy for the paper to be published once these are addressed.

Minor comments:

The conclusion 'SOA coatings did not affect the immersion ice nucleation ability of the dust particles in the temperature range 253 to 235 K irrespective of coating thickness (3 – 60 nm)' and similar statements in the abstract seem too strong. While most of the data does support the statement the very reasonable uncertainties in measured INAS suggest to me that there could be some effect that it is not possible to discern with certainty from the data. The data are certainly suggestive and surprising, I would have thought that SOA coating would make a measurable difference to the INAS of the dusts, but I do not think this study allows the conclusion that SOA will not 'impede or enhance the ice nucleation ability by immersion mode of mineral dust in the mixed-phase cloud regime'. The authors effectively acknowledge as much in their discussion but this subtlety is at least partially lost in the conclusions and abstract. Additionally, it is entirely conceivable that different results could be obtained if different dust samples were used. I do not think that two soil samples can be claimed to represent all desert dusts. To summarize, I think it should be made clearer that the study covers only a fairly narrow set of circumstances and that this topic likely needs further investigation. I do not think this detracts at all from the usefulness and interest of the study.

On a related note, it is increasingly clear that the ice nucleating ability of a mixed mineral dust depends on its composition, at least potentially (Harrison et al., 2016;Peckhaus et al., 2016). While the information is in the literature as stated I think there should be a table reporting mineralogy of the two samples.

Similarly, it is stated in the conclusions that fit to data in this work 'yield a parameterization for desert dusts', which seems a very general statement for measurements conducted on two samples. Also, it does not seem to me surprising that the INAS spectrum fits reasonably to that of Niemand et al. (2012) when the measured samples are two of the five or so used in Niemand et al. I would suggest removing both these

statements.

I am a little curious as to why this study was conducted on soil samples dug up from underground or collected from the surface. I would have thought either more directly atmospherically relevant samples or 'pure' mineral samples would be of greater interest. Possibly the authors think these samples are of substantial atmospheric relevance but I think this needs to be more thoroughly explained and justified if so. Finally, I realise it's the established name for this sample but is it really reasonable to call dust collected north of Cairo 'Saharan'?

Recognising it might be slightly sparse, I think the authors may want to consider a figure showing the absence of impact of thickness of coating on ice nucleation effectiveness. Currently, the reader is forced to refer back to table 1 to figure it out, which doesn't aid readability.

Specific comments:

Pg 15 Line 29- considered to be 'in' reasonable. What does reasonable mean? Some sort of quantitative description might be helpful, and perhaps a comment on how data produced by the different instrument types should be interpreted.

Pg 3 line 15- Ammonium sulphate has been observed to enhance ice nucleation of mineral dusts recently (Kumar et al., 2018;Whale et al., 2018). It may be appropriate to note this here.

Pg 4 line 23- Why is immersion freezing being mimicked? Is the process not immersion freezing?

P 7 line 31- Are convective clouds not natural?

Pg 10 Line 20- Why does the number of large particles change reported AF? A bit more discussion may help the reader.

Technical comments:

[Figure]

Section 4.4 follows section 3.3 currently, this should presumably be section 3.4.

Pg 1 Line 15- The first sentence of the abstract make it seem as if there were two sets of experiments conducted, which was not the case, I would suggest revising this.

Pg 2 line 28- 'inferred' should be 'by inference' or similar I think

Pg 3 line 20- I would use 'INPs' instead of INP. I would suggest checking that INP and INPS are used properly throughout.

Pg 7 line 33- pg 8 line 1- '...homogeneous temperature control of below ±0.3 K.' is clumsy.

Pg 8 line 29-32- contribution to aerosol number maybe?

Pg 10 line 16- 'Way above uncertainty' is a bit loose.

Pg 11 line 17- missing word after 'indicated'.

Pg 12 line 8- Favouring immersion freezing over what?

Pg 13 line 7- maybe mention the origin of this factor of 3.

Pg 15 line 20-21 'appreciable' is very vague.

Pg 16 line 6-8- This sentence is poorly written.

Reference list- Ullrich et al. 2017 and several others lack journal names and Megahead should be Megahed I believe. I suggest checking the list carefully.

References

Harrison, A. D., Whale, T. F., Carpenter, M. A., Holden, M. A., Neve, L., O'Sullivan, D., Vergara Temprado, J., and Murray, B. J.: Not all feldspars are equal: a survey of ice nucleating properties across the feldspar group of minerals, Atmos. Chem. Phys., 16, 10927-10940, 10.5194/acp-16-10927-2016, 2016. Kumar, A., Marcolli, C., Luo, B., and Peter, T.: Ice nucleation activity of silicates and aluminosilicates in pure water

and aqueous solutions – Part 1: The K-feldspar microcline, Atmos. Chem. Phys., 18, 7057-7079, 10.5194/acp-18-7057-2018, 2018. Peckhaus, A., Kiselev, A., Hiron, T., Ebert, M., and Leisner, T.: A comparative study of K-rich and Na/Ca-rich feldspar ice-nucleating particles in a nanoliter droplet freezing assay, Atmos. Chem. Phys., 16, 11477-11496, 10.5194/acp-16-11477-2016, 2016. Whale, T. F., Holden, M. A., Wilson, Theodore W., O'Sullivan, D., and Murray, B. J.: The enhancement and suppression of immersion mode heterogeneous ice-nucleation by solutes, Chemical Science, 9, 4142-4151, 10.1039/C7SC05421A, 2018.

---

## Referee Comment (RC2) · Anonymous Referee #2 · 5 Dec 2018

This study is a laboratory experiment measuring the ice-nucleating properties of mineral dusts coated with SOA, as determined by 3 separate types of instruments, the PINC, CFDCs, and the AIDA cloud chamber. The experiments appear to be well designed and meticulously carried out, and the redundancy of multiple ice nucleation instruments, while not necessary, strengthens one's confidence in the results. There are some differences between the analyses of data from the separate instruments, and also the interpretation of these results in light of previous work which require further clarifications, as discussed below. Once resolved, this paper will make a good contribution to the literature.

The topic of deactivation of ice nucleation activity by coatings has been reported in a number of previous manuscripts. The main conclusion from this paper is that SOA-type coatings do not deactivate the INP activity of 2 types of representative mineral dusts. This is opposite to the previous conclusions of many papers that report that deactivation does occur. However, this study is specific to measurements in the mixed phase cloud regime.

**Major Comments:**

1. The most interesting question in the manuscript is WHY does deactivation occur in some studies and not this one? My major comment is that this question warrants more attention and more organized discussion than is included in the current manuscript. References are made to other studies here and there throughout the text, but these are not summarized or reported in the context of all other studies making it different to draw any overarching conclusions. A systematic analysis or at least discussion of what does and does not lead to activation is needed.

A. A thorough discussion on the thicknesses of coating here compared to those in papers which did observe deactivation would greatly strengthen this paper. (Of course, such a discussion is dependent on availability of coating thickness (or aerosol mass/density changes such as those obtainable with a particle mass analyzer (PAM)) observations. The manuscript does conclude that no effect of coating thickness was observed over the range of 3-60 nm. However, it is possible that other studies were conducted on even thicker coatings. Is this known?

B. The multiple measurements (PINC, CFDC, AIDA) are in general agreement, which suggests that the reported fractions frozen and ice nucleation active site densities are relatively accurate. It follows that the reasons that deactivation of INP efficiency was not observed is related to sample generation. Are the substrate aerosols similar in size distributions to previous measurements? How does coating method compare to previous studies?

C. It may well be that some aerosols are more receptive to coatings that others, for either chemical or physical reasons. Chemically, acidity will vary with atmospheric coating compositions. Also, the sticking coefficient on a Teflon aerosol (for example) would be much lower than dust (note that this would not explain why the dust doesn't deactivate. Physically, it is possibly that highly irregular shaped particles and/or highly porous particles may be more difficult to coat and therefore less likely to deactivate as INP.

D. Are there any other reports of coatings NOT activating mineral dust INP?

E. Might deactivation of heterogeneous nucleation occur only for certain ice nucleation mechanisms not explored here? That too, would be interesting. For example, Sullivan (2010) observed very different effects of coating for super and subsaturated conditions, as mentioned on page. 3. The Sullivan

immersion freezing results do show deactivation, which is different than the results here. How do the temperatures of the 2 studies compare?

The manuscript currently takes a broad brush on activation vs. deactivation (pg. 3 ln 15…" the effects of inorganic acid and organic coatings on a variety of mineral dust particles" are all reported in one lump statement. It would be interesting to more carefully consider how variations in substrate aerosol result in more/less deactivation and also how differences in coating compositions lead to different results.

2. Pg 4 ln 28 Why does the CFDC require such a high supersaturation (105%) to simulate immersion freezing? Also, for reference, the PINC operating supersaturation should be reported at the same point in the text. Later, it is reported that the PINC's droplet survival region is at 107% ss and higher. Was 107% the ss chosen for immersion measurements?

 Further, I am confused about this survival statement- if droplets only survive at 107% and wetter, then how is all the Figure 6 and Figure 7 PINC data (at 105%) obtained?

3. Figure 8 and text page 13: The text says there is only one outlier below the 1:1 region on the figure. I see at least 2 outliers, one PINC and one CFDC.

4. pg 15. The conclusion that "observations of scatter between the 3 INP chambers can be attributed to differences in the evaluation of immersion…" appears to have been added as an afterthought. This is an important point and should be made and elaborated on earlier in the text.

**Minor Comments**

1. page 1 line 30. The last sentence in the abstract is grammatically incorrect. Revise.

2. page 4, ln 17. "There are no other studies in the MPC regime" I had to look back to find MPC defined. It should be written out. Also, since this statement is so central to the paper, the regime should be specified here, add "…that is, over the temperature range of …., and above water supersaturation range of …"(these values are currently provided later in the manuscript.)

2. page 15, ln 7, there is a misplaced phrase, (with T in C). Please revise sentence.

3. page 16, ln 6, "We note.." This is a run-on sentence that needs to be revised.

---

## Author Comment (AC1) · 30 Mar 2019

Reviewer comments in bold and authors' response in regular typeface.

**General comments:**

**This article reports that coating two types of mineral dust particles with a secondary organic aerosol proxy produced by dark ozonolysis of alpha-pinene made little difference to their immersion mode ice nucleating ability between 253 K and 233 K. No systematic differences in ice nucleating abilities of the dusts were observed with various atmospherically plausible coating thicknesses. Measurements were conducted simultaneously on three well established cloud chamber instruments with broadly consistent results. The study is relevant to the scope of ACP, presents useful and interesting new data and is scientifically sound. The paper is mostly well written. I have a few minor comments, but am quite happy for the paper to be published once these are addressed.**

We thank the reviewer for the positive comments and address the minor comments below.

**Minor comments:**

**The conclusion 'SOA coatings did not affect the immersion ice nucleation ability of the dust particles in the temperature range 253 to 235 K irrespective of coating thickness (3 – 60 nm)' and similar statements in the abstract seem too strong. While most of the data does support the statement the very reasonable uncertainties in measured INAS suggest to me that there could be some effect that it is not possible to discern with certainty from the data. The data are certainly suggestive and surprising, I would have thought that SOA coating would make a measurable difference to the INAS of the dusts, but I do not think this study allows the conclusion that SOA will not 'impede or enhance the ice nucleation ability by immersion mode of mineral dust in the mixed phase cloud regime'. The authors effectively acknowledge as much in their discussion but this subtlety is at least partially lost in the conclusions and abstract. Additionally, it is entirely conceivable that different results could be obtained if different dust samples were used. I do not think that two soil samples can be claimed to represent all desert dusts. To summarize, I think it should be made clearer that the study covers only a fairly narrow set of circumstances and that this topic likely needs further investigation. I do not think this detracts at all from the usefulness and interest of the study.**

This is a valid point and we now refer to this point both in the Abstract and Conclusions sections. In the abstract, we specifically removed the reference to the atmosphere and the mixed-phase cloud regime. In addition, despite the uncertainties from *INAS* densities, even for the *AF* results, it is clear that the coatings do not make a difference. As such, we specify that the lack of impeding or enhancing is specifically for the SOA coatings used in this study (page 1 line 31-32).

However, we include now in the paper a comprehensive discussion of how coatings impair deposition mode ice nucleation but do not impair (or only partially impair) immersion mode ice nucleation. The results from this work, combined with previous numerous studies of organic and inorganic coatings on a variety of dust particles suggest that organic coatings and inorganic coatings (depending on temperature regime) will not impair immersion mode ice nucleation activity. We now note this in the conclusions section (page 17 line 27 – page 18 line 3). We also acknowledge that we use a limited number of samples in this study in the conclusions section (page 17 lines 27-28).

**On a related note, it is increasingly clear that the ice nucleating ability of a mixed mineral dust depends on its composition, at least potentially (Harrison et al., 2016;Peckhaus et al.,**

**2016). While the information is in the literature as stated I think there should be a table reporting mineralogy of the two samples.**
This is indeed a reasonable suggestion, however the mineralogy is only available qualitatively for our SD sample in Linke et al. (2006, called Cairo2 in their paper). Here we give the percentages for our AD sample, but for the SD sample, we can only say that the clays and dolomite dominate the composition compared to feldspars, as was reported in (Linke et al. 2006). In addition the AD sample has its complete mineralogy reported with mass percentages in Boose et al. (2016, called Taklamakan in that paper). Since we do not have the mass percentages for the SD sample, we refrain from constructing a table to report the mineralogy, and instead discuss it in the text (page 16 line 28 to page 17 line 6).

**Similarly, it is stated in the conclusions that fit to data in this work 'yield a parameterization for desert dusts', which seems a very general statement for measurements conducted on two samples. Also, it does not seem to me surprising that the INAS spectrum fits reasonably to that of Niemand et al. (2012) when the measured samples are two of the five or so used in Niemand et al. I would suggest removing both these statements.**
We agree with the reviewer's comment that we have only presented two samples in this work. As such we remove the statement about the parameterization and instead say we obtained a fit for the desert dusts used in this work (page 18 line 12).

Furthermore given the technique for ice nucleation used in Niemand et al. (2012), was also one of the three techniques used in this work, we also removed the statement *"..compared well (to within a factor of 5) in the temperature range 254 – 232 K to a previously proposed parameterization for desert dust.."* from the conclusions section.

**I am a little curious as to why this study was conducted on soil samples dug up from underground or collected from the surface. I would have thought either more directly atmospherically relevant samples or 'pure' mineral samples would be of greater interest. Possibly the authors think these samples are of substantial atmospheric relevance but I think this needs to be more thoroughly explained and justified if so. Finally, I realise it's the established name for this sample but is it really reasonable to call dust collected north of Cairo 'Saharan'?**
Obtaining samples directly from airborne dust requires long collection times with high volume cyclone samplers necessitating proximity to the source for ground based sampling as was done for example in (Boose et al. 2016). Even in this work, the sampling conducted directly in a dust storm/event required sampling for over 24 hours and yielded enough sample on the order of 1-2 g to conduct one day of ice nucleation measurements. It would therefore not be possible to run multiple experiments with the three different instruments and various coating experiments because of the small sample size. We now clarify this on page 6 lines 17-19.

These samples were chosen because the Saharan and Asian regions are known to be the largest contributors to airborne atmospheric mineral dust from arid and semi-arid regions (Tang et al. 2016), we now clarify this point to motivate our sample choice better on page 6 lines 10-11.

We appreciate the reviewer's concern over naming of the sample. Depending on the source, all of Egypt is considered to be part of the Sahara, except the Nile valley, Nile delta and the region close to the Mediterranean coast. But even these descriptions depend on source. For example, the Encyclopedia Britannica reports that all of Egypt is part of Sahara and that Cairo and the

regions north of it are also within the borders of the Sahara arid zone. However, the around Cairo may not be a sand area (see figure below).

[Figure]

*1. Borders of the Saharan Region (taken from* [https://www.britannica.com/place/Sahara-desert-Africa/media/516375/200](https://www.britannica.com/place/Sahara-desert-Africa/media/516375/200))

**Recognizing it might be slightly sparse, I think the authors may want to consider a figure showing the absence of impact of thickness of coating on ice nucleation effectiveness. Currently, the reader is forced to refer back to table 1 to figure it out, which doesn't aid readability.**

This is a good idea. We have prepared a new figure to also show the impact of coating thickness (or lack thereof) on the ice nucleation properties on both the AD and SD samples. These have been added as Figure 6 in the revised manuscript and we refer to these on page 11 line 22, page 12 lines 20, 27 page 13 line 4, page 17 line 26. We retain the other figures as well in order to refer to the discussion on specific experiment numbers as well as to allow a reader to refer to the specific coating thickness reported in table 1.

**Specific comments:**

**Pg 15 Line 29- considered to be 'in' reasonable. What does reasonable mean? Some sort of quantitative description might be helpful, and perhaps a comment on how data produced by the different instrument types should be interpreted.**

We changed the word "reasonable" to "good" and clarified this means within overlapping uncertainties (page 18 line 9). A comment on how the data from different instruments is to be interpreted is already provided in the conclusions section on page 18 lines 3-9.

**Pg 3 line 15- Ammonium sulphate has been observed to enhance ice nucleation of mineral dusts recently (Kumar et al., 2018; Whale et al., 2018). It may be appropriate to note this here.**

We have now added that in addition to suppression of ice nucleation, enhancements are also observed for immersion freezing and cited the suggested studies and more (see page 4 line 5-13).

**Pg 4 line 23- Why is immersion freezing being mimicked? Is the process not immersion freezing?**

Here we meant to say "simulate" the atmospheric process of immersion freezing in the instruments we use. We now clarified this (see page 5 line 22).

**P 7 line 31- Are convective clouds not natural?**
We have corrected this. We meant to say "covering a range of weakly to strongly convective wave clouds" (page 9 line 6)

**Pg 10 Line 20- Why does the number of large particles change reported AF? A bit more discussion may help the reader.**
We agree, we now explained this further, by stating that larger particles are more effective INPs thus contribute to the INP population and will influence the *AF* if excluded from being sampled (see page 11 line 32 to page 12 line 3).

**Technical comments:**
**Section 4.4 follows section 3.3 currently, this should presumably be section 3.4.**
Thanks for catching that, now corrected.

**Pg 1 Line 15- The first sentence of the abstract make it seem as if there were two sets of experiments conducted, which was not the case, I would suggest revising this.**
Two sets of experiments were indeed conducted, one with coated and uncoated AD, the second with coated and uncoated SD. As such, we leave the sentence as is.

**Pg 2 line 28- 'inferred' should be 'by inference' or similar I think**
We changed 'inferred' to 'by inference' (page 2 line 28)

**Pg 3 line 20- I would use 'INPs' instead of INP. I would suggest checking that INP and INPS are used properly throughout.**
The change to INPs was done (now page 4 line 23). We also checked the whole manuscript and made a number of corrections to "INPs" and "an INP"

**Pg 7 line 33- pg 8 line 1- '…homogeneous temperature control of below _0.3 K.' is clumsy.**
We now changed this to read "*homogeneous temperature control < ±0.3 K*" (page 9, line 8)

**Pg 8 line 29-32- contribution to aerosol number maybe?**
It should be contribution to surface area, not aerosol number like suggested by the reviewer. However, the sentence was a little confusing, and thus we have clarified this to read "*However, the contribution to the surface area from the particles above 1 μm (see Figure 2) is significant enough (see Figure 3) to have to account for the impactors used upstream of PINC and CSU-CFDC in determining INAS densities*" (see page 10 line 5-7)

**Pg 10 line 16- 'Way above uncertainty' is a bit loose.**
We changed this to 'much greater than the maximum uncertainty in *AF* (28%)..' (page 11 line 28-29).

**Pg 11 line 17- missing word after 'indicated'.**
The word "by" was missing. Now corrected (page 12 line 33)

**Pg 12 line 8- Favouring immersion freezing over what?**

Favouring immersion freezing compared to deposition nucleation where coatings suppress the ice nucleation activity of dust INPs. This clarification has now been added (now page 14 line 5-7)

**Pg 13 line 7- maybe mention the origin of this factor of 3.**
The origin of the factor 3 has been discussed already in the introduction (page 5, line 23-28) and in the results and discussion section (page 10, lines 20-25) and thus at the said location (now page 15 line 11) we refer the reader to these sections rather than repeating the origin of the factor 3 yet again.

**Pg 15 line 20-21 'appreciable' is very vague.**
We have now clarified that by appreciable we mean tens of nm (page 17 line 26)

**Pg 16 line 6-8- This sentence is poorly written.**
We have now clarified this sentence (page 18 lines 15-17)

**Reference list- Ullrich et al. 2017 and several others lack journal names and Megahead should be Megahed I believe. I suggest checking the list carefully.**
Thanks for carefully checking this. Megahed now corrected and all references to have been checked to include journal names.

**References**
**Harrison, A. D., Whale, T. F., Carpenter, M. A., Holden, M. A., Neve, L., O'Sullivan, D., Vergara Temprado, J., and Murray, B. J.: Not all feldspars are equal: a survey of ice nucleating properties across the feldspar group of minerals, Atmos. Chem. Phys., 16, 10927-10940, 10.5194/acp-16-10927-2016, 2016.**

**Kumar, A., Marcolli, C., Luo, B., and Peter, T.: Ice nucleation activity of silicates and aluminosilicates in pure water and aqueous solutions – Part 1: The K-feldspar microcline, Atmos. Chem. Phys., 18, 7057-7079, 10.5194/acp-18-7057-2018, 2018.**

**Peckhaus, A., Kiselev, A., Hiron,T., Ebert, M., and Leisner, T.: A comparative study of K-rich and Na/Ca-rich feldspar ice-nucleating particles in a nanoliter droplet freezing assay, Atmos. Chem. Phys., 16, 11477-11496, 10.5194/acp-16-11477-2016, 2016.**
**Whale, T. F., Holden, M. A., Wilson, Theodore W., O'Sullivan, D., and Murray, B. J.: The enhancement and suppression of immersion mode heterogeneous ice-nucleation by solutes, Chemical Science, 9, 4142-4151, 10.1039/C7SC05421A, 2018.**

References

Boose, Y., et al. (2016), 'Heterogeneous ice nucleation on dust particles sourced from nine deserts worldwide – Part 1: Immersion freezing', *Atmos. Chem. Phys.,* 16 (23), 15075-95.
Linke, C., et al. (2006), 'Optical properties and mineralogical composition of different Saharan mineral dust samples: a laboratory study', *Atmospheric Chemistry and Physics,* 6, 3315-23.
Niemand, M., et al. (2012), 'A Particle-Surface-Area-Based Parameterization of Immersion Freezing on Desert Dust Particles', *Journal of the Atmospheric Sciences,* 69 (10), 3077-92.

Tang, M., Cziczo, D. J., and Grassian, V. H. (2016), 'Interactions of Water with Mineral Dust Aerosol: Water Adsorption, Hygroscopicity, Cloud Condensation, and Ice Nucleation', *Chem Rev,* 116 (7), 4205-59.

---

## Author Comment (AC2) · 30 Mar 2019

Reviewer comments in bold and authors' response in regular typeface.

**This study is a laboratory experiment measuring the ice-nucleating properties of mineral dusts coated with SOA, as determined by 3 separate types of instruments, the PINC, CFDCs, and the AIDA cloud chamber. The experiments appear to be well designed and meticulously carried out, and the redundancy of multiple ice nucleation instruments, while not necessary, strengthens one's confidence in the results. There are some differences between the analyses of data from the separate instruments, and also the interpretation of these results in light of previous work which require further clarifications, as discussed below. Once resolved, this paper will make a good contribution to the literature.**

**The topic of deactivation of ice nucleation activity by coatings has been reported in a number of previous manuscripts. The main conclusion from this paper is that SOA-type coatings do not deactivate the INP activity of 2 types of representative mineral dusts. This is opposite to the previous conclusions of many papers that report that deactivation does occur. However, this study is specific to measurements in the mixed phase cloud regime.**

The authors thank the reviewer for their comments

**Major Comments:**

**1. The most interesting question in the manuscript is WHY does deactivation occur in some studies and not this one? My major comment is that this question warrants more attention and more organized discussion than is included in the current manuscript. References are made to other studies here and there throughout the text, but these are not summarized or reported in the context of all other studies making it different to draw any overarching conclusions. A systematic analysis or at least discussion of what does and does not lead to activation is needed.**

This is a valid question. We have now extended the introduction to discuss deactivation (see page 3 line 1 to page 4 line 14) of INPs due to coatings. We discuss the differences in coating composition, aerosol substrate type and ice nucleation regime (deposition vs. immersion). Furthermore we discuss our results in this context in section 3.2 specifically page 12 line 18-22, page 13 line 3-13, and page 14 line 30 to page 15 line 2.

To answer the reviewer's question: The distinction between this study and previous ones that have reported deactivation has been the mode of ice nucleation and the composition of coating. For a study where deactivation of dust particles was observed due to the same type of coating as used here (Möhler et al., 2008), the mode of ice nucleation was different (deposition mode) vs immersion mode in this work, as was the temperature regime.

Previous studies have observed a reduction in ice nucleation activity in the immersion mode but to a lesser degree than in deposition mode, unlike the conclusion we draw here. These studies used $H_2SO_4$ for coating which was capable of modifying and irreversibly changing the surface of the mineral dust particles used (Sullivan et al., 2010b) and producing new reaction products on the surface (Sihvonen et al., 2014). On the other hand, coatings of levoglucosan on kaolinite particles also showed no affect on the ice nucleation activities in the immersion mode (Tobo et al., 2012). As such, the fact that we observe no deactivation of the INPs in the immersion mode in this work, is in agreement with previously reported literature. Coatings can either react and chemically change the ice nucleating surface as has been observed with inorganic acid coatings, or block access of water vapour to active sites in deposition mode, which then become exposed with the

coating is solvated in the immersion mode. This has been observed for dust particles coated with HNO₃ (Sullivan et al., 2010a; Kulkarni et al., 2015) which did not impede immersion mode freezing, similar to our results here. This aspect is discussed in the revised manuscript on page 14 line 2-17, page 14 line 30 to page 15 line 2.

**A. A thorough discussion on the thicknesses of coating here compared to those in papers which did observe deactivation would greatly strengthen this paper. (Of course, such a discussion is dependent on availability of coating thickness (or aerosol mass/density W changes such as those obtainable with a particle mass analyzer (PAM)) observations.**
We do not have PAM (or CPMA) measurements from the measurement period, however coating thicknesses were calculated using SOA yields and the surface area of aerosol in the chamber at the time of in-situ coating (see section 2.1 page 7 line 28 to page 7 line 8).

We have extended the discussion on coating thickness, and included new plots to demonstrate the lack of effect of coating thickness on the *INAS* densities (see Figure 6). The comparison of coating thickness in this work to that in the published literature is presented in section 3.2 specifically on page 12 line 18-22, page 12 line 36 to page 13 lines 13.

**The manuscript does conclude that no effect of coating thickness was observed over the range of 3-60 nm. However, it is possible that other studies were conducted on even thicker coatings. Is this known?**
This is a good point. The studies we compared our work to in the immersion mode, all have thinner coatings than the maximum reported in this work (60 nm). In particular for the studies that reported a partial or complete deactivation in the immersion mode freezing of dust particles due to coatings of H₂SO₄ (Sullivan et al., 2010b; Tobo et al., 2012; Augustin-Bauditz et al., 2014) looked at coating thicknesses between 1-15 nm. In particular Augustin-Bauditz et al. (2014) saw an influence on immersion mode activity with the thicker H₂SO₄ coating (15 nm) resulting in lower frozen fractions than feldspar with thinner H₂SO₄ coating (3 nm). Previous to this, Sullivan et al. (2010b) also found significant lowering of the *AF* in the immersion mode when coating thickness increased progressively from 1, 2.4 and 4.1 nm with the *AF* for the latter coating thickness being at the edge of the limit of quantification of the *AF*. On the other hand, studies that did not observe an effect of coating also do not report an effect of coating thickness. For example Kulkarni et al. (2014) report no effect of 1 and 40 nm thick H₂SO₄ coating on a variety of mineral dust particles in the immersion mode.

We note that the coatings discussed in the studies above are produced by transiting particles through heated vapour regions of the coating substance. As such thicker coatings are associated with higher processing temperature which could have an effect on the ice nucleation activity by enhancing reactions between the coating and the substrate particles. Sullivan et al. (2010b), showed that passing H₂SO₄ coated ATD particles through a 250 °C thermodenuder further reduces their ice nucleation activity compared to particles with the same coating thickness bypassing the thermodenuder.

We already discuss the coating thickness for the same type of SOA used here on illite and ATD particles from another study (Möhler et al., 2008) on page 14 lines 28-30. We now expand on this

by discussing the effect of coating thickness as described above (see section 3.2, page 12 line 18-22, page 12 line 26 to page 13 lines 13).

**B. The multiple measurements (PINC, CFDC, AIDA) are in general agreement, which suggests that the reported fractions frozen and ice nucleation active site densities are relatively accurate. It follows that the reasons that deactivation of INP efficiency was not observed is related to sample generation.**

This could be true, however, other important considerations should be noted. The regime of ice nucleation (deposition vs. immersion), or the physical and chemical properties of the coating itself, how reactive it is or is not with the dust surfaces. Prompted by the reviewer comments, we have discussed this in the introduction of the revised manuscript (see page 3 line 1 to page 4 line 14). Furthermore, we have discussed and now elaborated in section 3.2 on the role of sample generation (page 13 line 20-22, page 14 line 7-11), coating type and ice nucleation regime (page 14 line 5 – 7, page 14 line 19 to page 15 line 2).

**Are the substrate aerosols similar in size distributions to previous measurements? How does coating method compare to previous studies?**

Size of particles used in this study are predominantly in the submicron size range similar to or overlapping the range (250 – 700 nm) of those reported in previous literature and the studies we compare to in the manuscript. The coating method used here conducted at room temperature and dry conditions, differs from other methods that use heating followed by condensation. The use of heat for coating is now discussed on page 13 lines 13-18.

**C. It may well be that some aerosols are more receptive to coatings that others, for either chemical or physical reasons. Chemically, acidity will vary with atmospheric coating compositions. Also, the sticking coefficent on a Teflon aerosol (for example) would be much lower than dust (note that this would not explain why the dust doesn't deactivate. Physically, it is possibly that highly irregular shaped particles and/or highly porous particles may be more difficult to coat and therefore less likely to deactivate as INP.**

We agree with the reviewer that we could have incomplete coatings of our particles. The difficulty in achieving complete coatings and thus the lack of deactivation observed is acknowledged on page 13 line 20 and page 14 line 7. Given our coating procedure of producing excess amounts of SOA as is evident from the nucleation mode of aerosol produced, we expect that the coatings should be complete, but cannot rule out the possibility of incomplete coatings. Additionally we clarify that difficulty in achieving complete coatings can arise from the use of natural dust samples, which are expected to exhibit porous features and irregular shapes (see page 13 lines 20-21).

**D. Are there any other reports of coatings NOT activating mineral dust INP?**

We believe by stating "..coatings NOT activating mineral dust INP" the reviewer means "coatings not deactivating mineral dust"? If that is the case, then we have referred to such studies extensively in the introduction (specifically page 3 line 1 to page 4 line 14), and we now discuss both circumstances of coatings impeding and not impeding ice nucleation in section 3.1 (page 11 lines 7-18). Furthermore, we discuss other instances of coatings not deactivating ice nucleating particles in comparison to our results in section 3.2 page 13 lines 3-18 and the reasons with regard to the mode of ice nucleation and potential surface modification of the INP on page 14 line 30 to page 15 line 2. There are numerous reports of $H_2SO_4$ coatings not (fully) deactivating mineral dust in

the immersion mode, or reactive uptake of HNO$_3$ not deactivating immersion freezing of mineral dust at all. However, these studies often found that the deposition mode was deactivated for the same processes/coated aerosol. This points to a key difference between how the deposition and immersion modes respond to coatings and particle processing for mineral dust particles, as we discuss on page 3 line 1 to page 4 line 14, and in response to your question below.

**E. Might deactivation of heterogeneous nucleation occur only for certain ice nucleation mechanisms not explored here? That too, would be interesting. For example, Sullivan (2010) observed very different effects of coating for super and subsaturated conditions, as mentioned on page. 3.**

This is true, the RH regime investigated can result in different effects of coating. Sullivan et al. (2010a); Sullivan et al. (2010b) observed a stronger deactivation of mineral dust in the deposition more versus the immersion mode for H$_2$SO$_4$ coatings, and for reactive uptake of HNO$_3$ vapor, only the deposition mode was impeded, but immersion mode ice nucleation was not affected. Furthermore, in the water subsaturated regime (deposition mode) for the same type of SOA used here, Möhler et al. (2008) reported a suppression of ice nucleation for illite and ATD coated particles. We already discuss this in section 3.2 (page 14 line 5-7 and line 19 to page 15 line 2) but now further clarify the comparison of the ice nucleation regimes (subsaturated vs. supersaturated).

**The Sullivan immersion freezing results do show deactivation, which is different than the results here. How do the temperatures of the 2 studies compare?**

The temperature regime is similar, but the coating substance is different. In the subsaturated regime, complete deactivation is observed for both H$_2$SO$_4$ and HNO$_3$, however, in the saturated regime (immersion freezing), partial deactivation is only observed for H$_2$SO$_4$ and no deactivation for HNO$_3$ coated dust. We already discuss this in the manuscript on page 3 lines 5 to 21 and also now include this in discussing our results in section 3.1 (specifically page 11 lines 7-15). The somewhat different strength of the response to H$_2$SO$_4$ versus HNO$_3$ processing may be due to the heated vapour source used for H$_2$SO$_4$ coatings, and/or different chemical reaction pathways with the mineral components accessible to HNO$_3$ versus H$_2$SO$_4$, as well as different solubilities of HNO$_3$ vs. H$_2$SO$_4$ reaction products.

**The manuscript currently takes a broad brush on activation vs. deactivation (pg. 3 ln 15…" the effects of inorganic acid and organic coatings on a variety of mineral dust particles" are all reported in one lump statement. It would be interesting to more carefully consider how variations in substrate aerosol result in more/less deactivation and also how differences in coating compositions lead to different results.**

The sentence being referred to by the reviewer (page 3 line 15 in original manuscript) is now on page 3 line 1 onwards in the revised manuscript) We expand on the discussion of inorganic (page 3 lines 1 to page 4 line 13) and organic coatings (page 5 lines 4-19). Not only do we discuss the effects of the coating composition on supressing ice nucleation but we also include in our discussion: enhanced ice nucleation due to chemical treatments and the effects of the aerosol substrate and ice nucleation regime (deposition vs. immersion).

**2. Pg 4 ln 28 Why does the CFDC require such a high supersaturation (105%) to simulate immersion freezing?**

There are a number of reasons for this. First, the need to be well above $RH_w = 100\%$, to ensure all the aerosol particles activate into droplets in the short (~5 second) residence time of particles in the CFDCs. Second, the reported $RH_w$ of 105% is only true for a defined narrow aerosol lamina within the chamber outside which the RH should be lower. Aerosol particles are drawn into the chamber sandwiched between two particle-free sheath flows in order to constrain the aerosol to the narrow lamina. However, due to non-ideal flow conditions, the aerosol layer can have a width that may extend beyond the defined region of the set point RH and can result in being exposed to a variation of up to ±1-3% $RH_w$ from the set point conditions (depending on the temperature applied and the ratio of aerosol to sheath flows used). Third, if the sample width increases even more or particles escape from this defined layer which has been reported to occur (DeMott et al., 2015; Garimella et al., 2017), the sample could be exposed to even lower $RH_w$ than the set point $RH_w$. As such operating at $RH_w=100\%$ would mean a fraction of particles may not be exposed to water saturated conditions, and thus exaggerating the $RH_w$ to 105% ensures all particles are exposed to supersaturated conditions favourable for forming cloud droplets and thus freezing by immersion. We refer to this aspect on page 5 line 22-26, page 10, line 9-25.

**Also, for reference, the PINC operating supersaturation should be reported at the same point in the text.**
The sentence being referred to by the reviewer (now page 5 line 25) in the introduction section, is citing/discussing work from a previous publication and not describing the operation RH conditions in this work. The latter has now been added to the methods section for both PINC (section 2.2, page 8 lines 4-7) and CSU-CFDC (section 2.3 page 8 line 28). The sampling conditions are also discussed at the beginning of the results section on page 10 lines 9-25. Lastly, the operating supersaturation of PINC and CSU-CFDC is also labelled in the figures where the data is introduced for *INAS* and *AF* (Figures 4 – 8).

**Later, it is reported that the PINC's droplet survival region is at 107% ss and higher. Was 107% the ss chosen for immersion measurements?**
No, the evaluation RH for the results presented here was $RH_w=105\%$, the same as the CSU-CFDC. This has now been clarified at the end of section 2.2 page 8 lines 4-7.

**Further, I am confused about this survival statement- if droplets only survive at 107% and wetter, then how is all the Figure 6 and Figure 7 PINC data (at 105%) obtained?**
The droplet survival relative humidity defines the point at which water droplets that form in PINC and don't freeze survive the evaporation region at the end of the chamber and thus also get sampled by the detector (OPC). In the OPC, ice crystals are distinguished from unactivated aerosol particles only by size. I.e. all particles entering PINC have $d < 0.7$ μm (see section 2.2). If the particles nucleate ice, these grow to larger than 3 μm. As such only by detecting the number of particles in the size bins larger than 3 μm, one can assess the total number of ice crystals (hence INP) without interference from unfrozen droplets. If the chamber (PINC) is operated at $RH_w \geq 107\%$ , the water drops that survive would also be sampled by the OPC and have overlapping sizes with those of the ice crystals, and thus we would no longer be able to trust the OPC signal to indicate only ice. This means that $RH_w = 107\%$ is the maximum operable RH for the temperature regime presented here (< 253 K). This condition does not exclude the possibility of water droplets forming inside the growth region of PINC, but it just ensures that water drops are not sampled at the exit of PINC

since we are interested only in the ice crystals at the exit of PINC. We have discussed this (and added some clarifications in the revised manuscript) on page 8 line 1-11.

**3. Figure 8 and text page 13: The text says there is only one outlier below the 1:1 region on the figure. I see at least 2 outliers, one PINC and one CFDC.**
Indeed, there is only one outlier that does not overlap with the 1:1 line in Figure 8. The confusion for the reader may arise from the fact that we specifically refer to the region where $INAS_{(coated)} > 10^{10}$ sites m$^{-2.}$ i.e. specifically on the $y$-axis. To clarify this, we now specify "$y$-axis" in addition to the previously stated "$INAS_{(coated)} > 10^{10}$ sites m$^{-2}$" (see page15 line 15).

**4. pg 15. The conclusion that "observations of scatter between the 3 INP chambers can be attributed to differences in the evaluation of immersion…" appears to have been added as an afterthought. This is an important point and should be made and elaborated on earlier in the text.**
We agree with the reviewer, and the evaluation of immersion freezing for CFDCs compared to other devices has been addressed and elaborated upon before the conclusions, for example at the end of the introduction section (page 5 lines 22-28) and in Section 3.0 (page 10 lines 16-25). Furthermore, we have extensively discussed the scatter between the data of the three chambers in section 3.3 (see page 15 line 15 to page 16 line 2).

**Minor Comments**
**1. page 1 line 30. The last sentence in the abstract is grammatically incorrect. Revise.**
We agree, the sentence was clunky, and have now revised it also based on Rev. 1 comments (page 1 line 30-32).

**2. page 4, ln 17. "There are no other studies in the MPC regime" I had to look back to find MPC defined. It should be written out. Also, since this statement is so central to the paper, the regime should be specified here, add "…that is, over the temperature range of …., and above water supersaturation range of …"(these values are currently provided later in the manuscript.)**
Since MPC is used at so many instances (> 10) in the paper, we keep the acronym in this sentence, however we do clarify with temperature and relative humidity values how we define the MPC regime (see page 5 line 16 and line 21).

**2. page 15, ln 7, there is a misplaced phrase, (with T in C). Please revise sentence.**
This is now corrected (page 17 line 19).

**3. page 16, ln 6, "We note.." This is a run-on sentence that needs to be revised.**
We have now revised this sentence extensively as it was also critiqued by Reviewer 1 (page 18 lines 21-23).

References

Augustin-Bauditz, S., Wex, H., Kanter, S., Ebert, M., Niedermeier, D., Stolz, F., Prager, A., and Stratmann, F.: The immersion mode ice nucleation behavior of mineral dusts: A comparison of different pure and surface modified dusts, Geophys. Res. Lett., 41, 7375-7382, doi:10.1002/2014GL061317, 2014.

DeMott, P. J., Prenni, A. J., McMeeking, G. R., Sullivan, R. C., Petters, M. D., Tobo, Y., Niemand, M., Moehler, O., Snider, J. R., Wang, Z., and Kreidenweis, S. M.: Integrating laboratory and field data to quantify the immersion freezing ice nucleation activity of mineral dust particles, 15, 393-409, 2015.

Garimella, S., Rothenberg, D. A., Wolf, M. J., David, R. O., Kanji, Z. A., Wang, C., Rösch, M., and Cziczo, D. J.: Uncertainty in counting ice nucleating particles with continuous flow diffusion chambers, Atmos. Chem. Phys., 17, 10855-10864, doi:10.5194/acp-17-10855-2017, 2017.

Kulkarni, G., Sanders, C., Zhang, K., Liu, X., and Zhao, C.: Ice nucleation of bare and sulfuric acid-coated mineral dust particles and implication for cloud properties, J. Geophys. Res.-Atmos., 119, 2014JD021567, doi:10.1002/2014JD021567, 2014.

Kulkarni, G., Zhang, K., Zhao, C., Nandasiri, M., Shutthanandan, V., Liu, X., Fast, J., and Berg, L.: Ice formation on nitric acid-coated dust particles: Laboratory and modeling studies, 120, 7682-7698, doi:10.1002/2014JD022637, 2015.

Möhler, O., Benz, S., Saathoff, H., Schnaiter, M., Wagner, R., Schneider, J., Walter, S., Ebert, V., and Wagner, S.: The effect of organic coating on the heterogeneous ice nucleation efficiency of mineral dust aerosols, Environ. Res. Lett., 3, 025007, doi:10.1088/1748-9326/3/2/025007, 2008.

Sihvonen, S. K., Schill, G. P., Lyktey, N. A., Veghte, D. P., Tolbert, M. A., and Freedman, M. A.: Chemical and Physical Transformations of Aluminosilicate Clay Minerals Due to Acid Treatment and Consequences for Heterogeneous Ice Nucleation, J. Phys. Chem. A, 118, 8787-8796, doi:10.1021/jp504846g, 2014.

Sullivan, R. C., Minambres, L., DeMott, P. J., Prenni, A. J., Carrico, C. M., Levin, E. J. T., and Kreidenweis, S. M.: Chemical processing does not always impair heterogeneous ice nucleation of mineral dust particles, Geophys. Res. Lett., 37, L24805, doi:10.1029/2010gl045540, 2010a.

Sullivan, R. C., Petters, M. D., DeMott, P. J., Kreidenweis, S. M., Wex, H., Niedermeier, D., Hartmann, S., Clauss, T., Stratmann, F., Reitz, P., Schneider, J., and Sierau, B.: Irreversible loss of ice nucleation active sites in mineral dust particles caused by sulphuric acid condensation, Atmos. Chem. Phys., 10, 11471-11487, doi:10.5194/acp-10-11471-2010, 2010b.

Tobo, Y., DeMott, P. J., Raddatz, M., Niedermeier, D., Hartmann, S., Kreidenweis, S. M., Stratmann, F., and Wex, H.: Impacts of chemical reactivity on ice nucleation of kaolinite particles: A case study of levoglucosan and sulfuric acid, Geophys. Res. Lett., 39, doi:L19803

10.1029/2012gl053007, 2012.